# Aberrant Expression and Prognostic Potential of IL-37 in Human Lung Adenocarcinoma

**DOI:** 10.3390/biomedicines10123037

**Published:** 2022-11-24

**Authors:** Panayiota Christodoulou, Theodora-Christina Kyriakou, Panagiotis Boutsikos, Maria Andreou, Yuan Ji, Damo Xu, Panagiotis Papageorgis, Maria-Ioanna Christodoulou

**Affiliations:** 1Tumor Immunology and Biomarkers Laboratory, Basic and Translational Cancer Research Center, Department of Life Sciences, School of Sciences, European University Cyprus, Nicosia 2404, Cyprus; 2School of Medicine, European University Cyprus, Nicosia 2404, Cyprus; 3Tumor Microenvironment, Metastasis and Experimental Therapeutics Laboratory, Basic and Translational Cancer Research Center, Department of Life Sciences, School of Sciences, European University Cyprus, Nicosia 2404, Cyprus; 4School of Infection and Immunity, University of Glasgow, Glasgow G12 8TA, UK; 5State Key Laboratory of Respiratory Disease for Allergy at Shenzhen University, Shenzhen Key Laboratory of Allergy and Immunology, Shenzhen University School of Medicine, Shenzhen 518055, China

**Keywords:** interleukin (IL-)37, lung adenocarcinoma, biomarker, survival, infiltration rates

## Abstract

Interleukin-37 (IL-37) is a relatively new IL-1 family cytokine that, due to its immunoregulatory properties, has lately gained increasing attention in basic and translational biomedical research. Emerging evidence supports the implication of this protein in any human disorder in which immune homeostasis is compromised, including cancer. The aim of this study was to explore the prognostic and/or diagnostic potential of IL-37 and its receptor SIGIRR (single immunoglobulin IL-1-related receptor) in human tumors. We utilized a series of bioinformatics tools and -omics datasets to unravel possible associations of IL-37 and SIGIRR expression levels and genetic aberrations with tumor development, histopathological parameters, distribution of tumor-infiltrating immune cells, and survival rates of patients. Our data revealed that amongst the 17 human malignancies investigated, IL-37 exhibits higher expression levels in tumors of lung adenocarcinoma (LUAD). Moreover, the expression profiles of IL-37 and SIGIRR are associated with LUAD development and tumor stage, whereas their high mRNA levels are favorable prognostic factors for the overall survival of patients. What is more, *IL-37* correlates positively with a LUAD-associated transcriptomic signature, and its nucleotide changes and expression levels are linked with distinct infiltration patterns of certain cell subsets known to control LUAD anti-tumor immune responses. Our data indicate the potential value of IL-37 and its receptor SIGIRR to serve as biomarkers and/or immune-checkpoint therapeutic targets for LUAD patients. Further, the data highlight the urgent need for further exploration of this cytokine and the underlying pathogenetic mechanisms to fully elucidate its implication in LUAD development and progression.

## 1. Introduction

Interleukin-37 (IL-37) is one of the latest members included in the IL-1 family of cytokines, known to suppress innate immune responses and modulate acquired ones. Thus, this cytokine possesses a pivotal role in inflammation related to the pathophysiology of various human disorders, including autoimmune diseases, inflammatory systemic conditions, infections, and cancer [1]. It is produced by immune and non-immune cells and acts via inhibition of the production of pro-inflammatory cytokines and activation of anti-inflammatory signals [2]. Similar to other immune-regulatory cytokines (e.g., TGF-β and IL-10), IL-37 has attracted notable interest both from a basic biological but also from a translational research perspective [2].

The human *IL-37* gene is located on chromosome 2q12-13, very close to the regulatory regions of the genes encoding the IL-1a and IL-1b cytokines [3]. The gene encodes for five protein isoforms, a, b, c, d, and e; however, their specific functions as well as their relative abundance are not yet fully elucidated [2]. Among all isoforms, IL-37b is the longest one (consisting of five of the six exons; all except exon 3) and the most well-studied [4,5]. In humans, IL-37 is reported to be constitutively expressed by circulating monocytes, tissue macrophages, dendritic cells (DCs), tonsil B cells, and plasma cells [2,6]. Upon pro-inflammatory stimuli, its expression is significantly augmented by tissue cells and peripheral blood mononuclear cells (PBMCs), predominantly monocytes, thus possessing a prevailing immunoregulatory role [1,6,7]. 

IL-37 exerts its effects on target cells via two distinct mechanisms: (i) Extracellularly, it binds to interleukin 18 receptor 1 (IL-18R1), which recruits interleukin 1 receptor 8 (IL-1R8, also known as single Ig IL-1-related receptor—SIGIRR), essential for the anti-inflammatory actions of the cytokine and forms a complex that transduces the signal intracellularly [8,9,10,11]; the activity of the cytokine lies mainly in the suppression of pro-inflammatory signaling factors, including mTOR (mammalian target of rapamycin), STAT1 (signal transducer and activator of transcription 1), AKT (Ak strain transforming), p53, p38, SHP-2 (SH2 domain-containing protein tyrosine phosphatase-2), Syk (Spleen Associated Tyrosine Kinase) [2], and also on the enhancement of anti-inflammatory signaling factors, such as PTEN (phosphatase and tensin homolog) phosphatase, to further inhibit inflammation through PI_3_K (Phosphoinositide 3-kinases) kinase, mTOR, MAPK (mitogen-activated protein kinase), and FADK (focal adhesion kinase) pathways [12]. (ii) Intracellularly, upon its cleavage by caspase-1 at aspartic acid (D20) residue and binding to Smad3, IL-37 can be translocated into the nucleus, where it dampens the expression of inflammatory genes [13,14,15,16].

Evidence of the role of IL-37 in cancer, emerging during the last years, support its tumor-protective properties exerted through the enhancement of anti-tumor immunity, specifically within the tumor microenvironment (TME). At this point, it is worthwhile to highlight the importance of the local microenvironment, consisting of distinct immune and non-immune, cellular and non-cellular components (growth factors, chemokines, cytokines), in the development and progression of human tumors, as well as response-to-treatment [17]. TME interplays are orchestrated by tumor-infiltrating lymphocytes (TILs), natural killer (NK) cells, tumor-infiltrating dendritic cells (TIDCs), tumor-associated macrophages (TAMs), tumor-associated neutrophils (TANs), cancer-associated fibroblasts (CAFs), and myeloid-derived suppressor cells (MDSCs). Within these cell types, T helper (Th) 1, cytotoxic T, NK, B cells, M1 macrophages (ΜΦ), and mature DCs represent partners of immune control against malignant cells, and Th2, regulatory T cells (Tregs), M2 ΜΦ, neutrophils, CAFs, immature DCs, and MDSCs promote immune escape. Essentially, certain elements of TME have been targeted by therapeutic drugs (antibodies against immune checkpoints, such as PD-1/PD-L1, as well as anti-angiogenic factors, such as anti-VEGF-A) that are associated with good clinical outcomes [18]. 

IL-37 overexpression by TAMs derived from patients with human hepatocellular carcinoma (HCC) inhibits M2 polarization via regulation of the IL-6/STAT3 pathway, to suppress tumor growth in vivo [19]. High IL-37 expression by HCC tumor cells is associated with upregulated levels of CCL3 and CCL20 and increased recruitment of CD1a^+^ dendritic cells (DCs) in tumor infiltrations [20]. What is more, IL-37 secreted from HCC cells enhances the expression of MHC-II, CD86, and CD40 surface molecules and the secretion of IL-2, IL-12, IL-12p70, interferon-a (IFN-α), and IFN-γ cytokines by DCs, which is, in turn, associated with an increased proportion of IFNγ^+^CD8^+^ T cells [20]. Additional in vivo experiments showed that overexpression of IL-37 in HCC cells resulted in increased recruitment of CD11c^+^ DCs in the tumor microenvironment and tumor growth delay [20]. On the other hand, a very recent study using an experimental colorectal cancer (CRC) model reported that IL-37 transgenic mice are highly prone to developing colitis-associated CRC, which is characterized by severely increased tumor burdens and dysfunction of infiltrating CD8^+^ T cells, dependent on SIGIRR [21]. 

Apart from immune-related effects, IL-37 also exerts its anti-cancer activity on other aspects of tumor development. First, it acts as an anti-angiogenic factor; its expression by cancer cells suppresses the tubule formation of human umbilical vein endothelial cells (HUVEC) cells in vitro, decreases the expression of matrix metallopeptidase 2 (MMP2) and vascular endothelial growth factor (VEGF) in SK-Hep-1 cells, and inhibits tumor angiogenesis in a murine model of HCC [22]. Second, it suppresses migration through the inhibition of Rac1 activation in various tumor cell types; indeed, intracellular IL-37 binds to the C-terminal region of the protein, preventing its membrane translocation and downstream signaling [22,23]. It has been observed that decreased expression of IL-37 in human lung adenocarcinoma (LUAD) biopsies is associated with tumor metastasis [23]. Lastly, the cytokine can act against tumor progression through the modulation of N6-methyladenosine (m6A) activity and inhibition of the Wnt5a/5b pathway in lung cancer cells [24].

Clinical observations over the last years have shed light on the potential of this cytokine to serve as a possible biomarker in various human malignancies. In CRC patients, serum IL-37 levels were found to be significantly elevated and positively correlated with the levels of CEA (carcinoembryonic antigen), a commonly used diagnostic biomarker for the disease [21]. In these patients, a negative correlation between IL-37 levels in the serum and CD8^+^ T cell infiltration in the tumor was also observed [21]. Importantly, IL-37 expression in CRC tumors was found (a) to be linearly correlated with their stage, with the highest expression detected in stage I and the lowest in stage IV tumors; and (b) to be associated with survival rates, with higher levels predicting longer disease-free (DFS) and overall (OS) survival [25]. It is of note that intratumoral IL-37 levels, together with the incidence of CD66b^+^ neutrophils, as well as mismatch repair (MMR) status, have been proven to be independent prognostic factors and are included in nomograms predicting DFS and OS in CRC, which could facilitate individualized patients’ management [25].

Elevated serum IL-37 levels were also detected in patients with transitional cell carcinoma of the bladder (TCC) [26]. In melanoma, high levels of IL-37 expressed by peripheral Tregs were found to mirror the secretion of IL-1β mediators, especially TGFβ, by the tumor, suggesting it could be used as a possible biomarker for tumor-induced immunosuppression [27]. Furthermore, in HCC tumor infiltration high prevalence of IL-37^+^CD1a^+^ DCs biopsies was linked to higher survival rates of patients [20]. On the other hand, the ratio of IL-18-to-IL-37 levels was higher in the serum and PBMCs of patients with oral squamous cell carcinoma (OSCC) compared to non-cancer individuals and associated with shorter OS and DFS [28]. Low levels of IL-37 in the sera of patients with acute myeloid leukemia (AML) were shown to be associated with poor prognosis of the disease, but they were restored to normal in complete remission [29]. Finally, in breast cancer, peripheral blood *IL-37* mRNA levels and CD8^+^ T cell numbers were decreased in patients compared to healthy individuals, and they were correlated with ER^+^/PR^+^/HER2^+^ status [30]. 

In this study, we aimed at the investigation of the possible prognostic potential of IL-37 in patients with cancer utilizing bioinformatics tools and publicly available databases. Since our initial results indicated that among various human malignancies, *IL-37* exerts its highest expression in lung adenocarcinoma (LUAD), the study was subsequently focused on this cancer type. IL-37 levels were found to be correlated with tumor development, stage, grade, and the improved overall survival of patients, and mutations and gene expression levels were associated with the differential distribution of immune cells infiltrating the tumor. 

## 2. Materials and Methods

### 2.1. Study Design

We first investigated the possible differential distribution of *IL-37* and *SIGIRR* expression levels in various human cancers using the Tissue Atlas tool of the Human Protein Atlas website [31]. Lung cancer, and more specifically lung adenocarcinoma (LUAD), was selected for further analysis. The TNMplot web tool [32] was used to compare the mRNA expression levels in LUAD versus non-LUAD specimens, the UALCAN portal [33] to explore the differential distribution among tumors of various histology, stage, nodal metastasis, or TP53 status, and the Kaplan–Meier plotter tool [34] to assess the effect of mRNA levels on survival rates of LUAD patients. Protein expression profiles as well as associations with various parameters of the pathology of the tumor were investigated through the UALCAN portal [33] and the Pathology tool of the Human Protein Atlas website [31]. To explore the expression distribution of *IL-37* and *SIGIRR* genes in different cell types of the human lung, the Single-Cell Type Atlas part of the Human Protein Atlas was used [35]. Finally, the effect of *IL-37* nucleotide changes or aberrations in expression levels on the differential distribution of various immune cell subsets infiltrating the LUAD tumor was analyzed using the TIMER2.0 webserver [36].

### 2.2. Study of the Expression Levels of IL-37 and SIGIRR in Various Human Cancers

To explore the expression patterns of *IL-37* and *SIGIRR* in various human cancers, the Tissue Atlas tool [31] of the Human Protein Atlas website was used (www.proteinatlas.org, accessed on 24 August 2022). The program processes data from RNA-sequencing experiments on tumor samples of various origin (17 different types of cancer, n = 7932 total samples). The mean of the FPKM levels of the genes in each cancer type and in the total cohort of cancer patients as well as its standard deviation (SD) was estimated. Non-parametric Mann–Whitney U test were applied for the evaluation of differences in gene expression levels.

### 2.3. Investigation of IL-37 and SIGIRR Expression Levels in LUAD versus Non-LUAD Lung Tissue

The TNMplot tool (www.tnmplot.com, accessed on 3 September 2022) [32] was used to explore impaired expression levels of the *IL-37* and *SIGIRR* genes in LUAD tumors. Comparative analysis processed RNA-sequencing data deposited in The Cancer Genome Atlas (TCGA) database from (a) 524 LUAD versus 486 non-LUAD individuals and (b) 57 pairs of LUAD versus adjacent normal tissue biopsies. Fold-changes of the median expression levels between the groups and the *p*-values assessed utilizing the non-parametric Mann–Whitney U test are reported; significant changes were considered those with a *p*-value < 0.05 and a fold-change > 2 or <0.5.

### 2.4. Exploration of Associations between IL-37 or SIGIRR Expression Levels with Certain Pathological Characteristics of the LUAD Tumor

To investigate possible associations between the gene-expression levels of *IL-37* and *SIGIRR* with certain parameters of the pathology of the tumor, data from the UALCAN portal were processed http://ualcan.path.uab.edu/ [33] (accessed on 3 September 2022). Gene expression levels were analyzed in correlation with histological type, cancer stage, nodal metastasis, and TP53 mutation status. Fold-changes compared to control groups >2 or <0.5 and *p*-values < 0.05 were considered significant.

### 2.5. Assessment of the Effect of IL-37 and SIGIRR Expression Levels on Survival Rates of Patients with LUAD

The Kaplan–Meier (KM) Plotter tool (www.kmplot.com, accessed on 24 August 2022) [34] was used to evaluate the ability of *IL37* expression to serve as prognostic factor for overall or relapse-free survival (OS or RFS, respectively) in LUAD patients. The tool analyzed RNA-sequencing data from 504 LUAD participants (deposited in Gene Expression Omnibus (GEO), EGA (European Genome-Phenome Archive), and TCGA databases) categorized based on *IL-37* or *SIGIRR* levels of expression ranging from high to low and calculated the hazard ratio (HR) and logrank *p*-values for the probability of survival at 250 months.

### 2.6. Development of a List of Positively/Negatively Correlated Genes of IL-37 in LUAD Biopsies

To obtain a list with genes whose expression levels are positively or negatively correlated with those of *IL-37*, the UALCAN tool (http://ualcan.path.uab.edu/ (accessed on 20 September 2022)) was utilized [33]. The tool processed RNAseq data from LUAD tumor biopsies of TCGA. A heatmap was generated, and Pearson’s correlation test was applied to assess the significance of the data (*p* < 0.05 were considered significant). Correlations of *IL-37* with genes encoding IC proteins were specifically explored via the Correlation analysis option of the TNM plotter tool [32]. Data from RNA seq experiments on LUAD tumors were processed; Spearman’s rho and *p*-values were obtained, and a correlation coefficient cut-off = 0 was set. 

### 2.7. Investigation of the IL-37 and SIGIRR Protein Expression Levels 

Protein expression in LUAD tumors was first explored using the Pathology tool of the Human Protein Atlas website, www.proteinatlas.org [35] (accessed on 20 September 2022). Deposited pictures of immunohistochemically stained sections of paraffin-embedded LUAD tissues were observed. Staining had been performed using polyclonal antibodies against human IL-37 (HPA054371) and SIGIRR (HPA023188). Levels of protein expression (z-values) in paired LUAD primary vs. normal tissues, as well as in tumors of different histological subtype, stage, grade, status of the HIPPO, WNT, mTOR, NRF2, RTK, or p53/Rb-related pathways, SWI-SNF complex, MYC/MYCN or chromatin modifier, were analyzed through the UALCAN web portal http://ualcan.path.uab.edu/ (accessed on 1 October 2021) [33]. *p*-Values calculated utilizing the non-parametric Mann–Whitney U test for differences in between-two group analyses are reported; *p*-values < 0.05 were considered significant.

### 2.8. Blood and Immune Single-Cell Analysis in Lung Tissue

To visualize single-cell RNA-seq (scRNAseq) data from human lung tissue, the Single-Cell Type Atlas was used. *IL-37* and *SIGIRR* expression profiles of blood and immune cells including macrophages, alveolar cells type 1 and 2, T cells, granulocytes, fibroblasts, club cells, ciliated cells, and endothelial cells are depicted in colored clusters at UMAP plots and in bar charts. Elevated expression levels (read counts normalized to transcripts per million protein coding genes, pTPM) of *IL-37* and *SIGIRR* in different blood and immune cell groups can categorize genes as cell type-enriched (at least four-fold higher mRNA level in a certain cell type compared to any other cell type), group-enriched (at least four-fold higher average mRNA level in a group of 2–10 cell types compared to any other cell type), and cell type-enhanced (at least four-fold higher mRNA level in a cell certain cell type compared to the average level in all other cancer types).

### 2.9. Analysis of Possible Correlations between IL-37 Gene Alterations or Expression Levels with Immune Cell Infiltration Patterns

To analyze the effect of *IL-37* nucleotide changes on or the association of *IL-37* expression levels with immune cell infiltration of lung tumors, the TIMER2.0 webserver was used [36]. The “mutation” module was utilized for the investigation of possible differential distribution of macrophages, CD4^+^ or CD8^+^ T cells, Tregs, dendritic cells (DCs), neutrophils, B cells, monocytes, NK cells, MDSCs, and endothelial cells in LUAD tumors of patients with *IL-37* somatic mutations vs. those without. The Wilcoxon *p*-value and log_2_fold-change of infiltration levels between the groups were estimated. The “gene” module was utilized for the exploration of possible correlations between the expression levels of *IL-37* and the levels of infiltration of the tumors by CD4^+^ or CD8^+^ T cells, Tregs, γδ T cells B cells, neutrophils, monocytes, macrophages, DCs, NK, mast cells, cancer-associated fibroblasts, lymphoid, myeloid or granulocyte-lymphocyte progenitor cells, endothelial cells, eosinophils, hematopoietic stem cells, and MDSCs. All data were filtered for tumor purity. Spearman’s rho and *p*-values were calculated for the evaluation of the linear positive or negative correlation. 

### 2.10. Analysis of Possible Correlations between IL-37 Gene Alterations or Expression Levels with Immune Cell Infiltration Patterns

The STRING (Search Tool for the Retrieval of Interacting Genes/Proteins) database (https://string-db.org) [37] (accessed on 1 October 2022) was used to explore the cancer/LUAD molecular/cellular networks in which IL-37 signaling is implicated. The Biological Process (Gene Ontology), Molecular Function (Gene Ontology), Cellular Component (Gene Ontology), Reference publications (PubMed), Local network cluster (STRING), KEGG Pathways, Reactome Pathways, WikiPathways, Tissue expression (TISSUES), Subcellular localization (COMPARTMENTS), Annotated Keywords (UniProt), Protein Domains (Pfam), Protein Domains and Features (InterPro), and Protein Domains (SMART) depositories were searched. The level of confidence for the minimum interaction score was 0.4. Pathways with a false-discovery rate (FDR) < 0.05 were considered to be significantly affected.

### 2.11. Statistical Analysis

Part of the statistical analysis of the data had been performed by the aforementioned web portals. Additional statistical tests included (as required): (a) for between two-groups analysis, non-parametric Mann–Whitney *U* or unpaired t-test with Welch’s correction; (b) for differences among means, an ordinary one-way ANOVA test; (c) for linear correlations, Pearson’s *r* test. The analysis was performed using the GraphPad Prism 8.4.2 software (GraphPad Software, San Diego, CA, USA). *p*-Values < 0.05 were considered significant.

## 3. Results

### 3.1. Among Human Cancers, IL-37 Exhibits the Highest Expression Levels in Lung Adenocarcinoma (LUAD)

The first aim of this study was to investigate the expression patterns of *IL-37* and its receptor, *SIGIRR,* in tumor biopsies of various human cancer types, utilizing the Tissue Atlas tool of the Human Protein Atlas database [31]. Analysis revealed that the *IL-37* expression pattern significantly differs in various types of human cancers (one-way ANOVA *p* < 0.0001), whereas among the 17 types examined, lung cancer exhibited the highest *IL-37* mRNA levels (mean ± SD: 6.5 ± 31.88, n = 994) (Table 1, Figure 1A). Lung cancer specimens were further analyzed based on their specific type: either lung adenocarcinoma (LUAD) or lung squamous carcinoma (LUSC); the former exhibited increased *IL-37* expression levels compared to the latter (12.85 ± 44.05, n = 500 vs. 0.25 ± 1.79, n = 494, respectively; Mann–Whitney *p* = 0.0001). In contrast, *SIGIRR* was expressed at similar levels in almost all cancer types, besides glioma and head-and-neck tumors, in which its expression levels were decreased compared to the rest (Table 1, Figure 1Β). However, within lung cancer samples, LUAD specimens exhibited higher *SIGIRR* levels compared to LUSC specimens (41.6 ± 4.46 vs. 13.64 ± 7.03, respectively; Mann-Whitney *p* < 0.0001). It is also noteworthy that, overall, all human tumor biopsies express lower *IL-37* compared to *SIGIRR* mRNA levels (1.029 ± 11.68 vs. 13.04 ± 8.911, respectively, n = 7932, *p* < 0.0001).

### 3.2. IL-37 Levels Are Increased in LUAD versus Non-LUAD Lung Tissues

Following the observation that LUAD exhibits the highest levels of *IL-37* expression among solid tumors, we investigated its differential expression pattern in cancerous (LUAD) vs. non-cancerous lung biopsies (non-LUAD) using the TNMplot web tool [32]. *IL-37* mRNA levels were found to be increased by 12-fold in samples from LUAD (n = 524) compared to those from non-LUAD individuals (n = 486, Mann–Whitney *p* = 3.83 × 10^−59^), and by 11-fold in tumor compared to paired adjacent normal tissues from LUAD individuals (n = 57; *p* = 3.81 × 10^−8^) (Figure 2A). *SIGIRR* mRNA expression was slightly decreased (fold-change = 0.85; *p* = 5.63 × 10^−5^) between LUAD and non-LUAD individuals; however, no change was observed between tumor and adjacent normal tissues in LUAD patients (fold-change = 1.05; *p* = 6.28 × 10^−1^) (Figure 2A).

### 3.3. IL-37 and SIGIRR Levels Are Associated with Histological Type and Tumor Grade of LUAD Tumors

Based on the UALCAN webtool analysis [33], *IL-37* and *SIGIRR* expression levels derived from RNA-sequencing experiments are associated with the histological type of LUAD tumors. The one-way ANOVA test revealed significant differential expression among groups (*p* = 0.0001). More specifically, LBC (lung bronchioloalveolar carcinoma)-mucinous and mucinous-coloid tumors express significantly higher *IL-37* levels (TPM median (range), fold-change, *p*-value, respectively; for LBC-mucinous (n = 5): 1.07 (0.20–1.96), 13.38, 0.0072 and for mucinous-coloid (n = 10): 0.26 (0–0.76), 3.25, 0.0153) compared to normal lung tissue (n = 59, TPM median (range) 0.08 (0–0.38)) (Figure 2B, Table 2). Regarding *SIGIRR* levels, the observed changes were modest (fold-changes: between 0.5–2; one-way ANOVA *p* > 0.05). *IL-37* levels also correlated with LUAD tumor stage. Compared to controls, stage 1 biopsies (n = 277) expressed 7.63 times higher mRNA levels (TPM median (range) = 28 (0–23.62)) (Figure 2B, Table 2). Compared to stage 1 biopsies, expression levels were significantly decreased in stage 2 (n = 125; 0.37 (0–13.55)) and stage 4 (n = 28; 0.35 (0–6.17)) specimens, being 4.63- and 4.38-fold, respectively, higher than controls. In stage 3, *IL-37* levels were 2.13-fold increased compared to non-LUAD biopsies (n = 85; 0.17 (0–19.55)). A notable trend of association was observed between *SIGIRR* levels and the stage of the LUAD tumor; none of the stage subgroups exhibited significant change compared to controls (n = 59; 35.06 (17.01–55.55)) (fold-changes ranged between 1.21 and 0.73). However, *SIGIRR* levels displayed a statistically significant linear decrease among subgroups, following the stage 1-to-4 order (one-way ANOVA test: *r*^2^ = 0.6909, *p* < 0.0001). Between-group analyses revealed that *SIGIRR* levels were significantly lower in biopsies of stage 2 compared to those of stage 1 (36.16 (12.59–75.62) vs. 42.53 (3.97–89.73), unpaired t-test *p* < 0.0001) in biopsies of stage 3 (33.03 (3.84–74.01)) compared to those of stage 1 (*p* < 0.0001), in biopsies of stage 3 compared to those of stage 2 (*p* = 0.038), and in biopsies of stage 4 (25.65 (7.76–69.02)) compared to those of stage 2 (*p* = 0.022). When compared to the control group, only stage 1 biopsies were found to express significantly different (higher) *SIGIRR* levels (*p* < 0.0001). 

**Figure 2 biomedicines-10-03037-f002:**
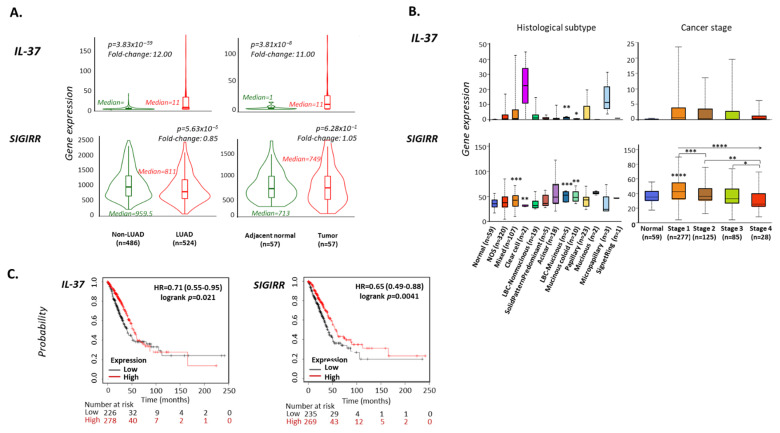
(**A**) Violin plots depicting the differential expression levels of *IL-37* and *SIGIRR* in lung tissues from LUAD (n = 486) vs. non-LUAD (n = 524) individuals or paired tumor vs. adjacent normal tissues from LUAD patients (n = 57 pairs). Median expression levels, Mann–Whitney *p*-values, and fold-changes between medians are reported. Data were obtained from www.tnmplot.com [32] (accessed on 3 September 2022). (**B**) Box and Tukey whiskers diagrams showing the differential distribution of *IL-37* and *SIGIRR* expression levels (FPKMs) as analyzed by RNA sequencing in LUAD samples of different histological type or stage. Data were obtained from http://ualcan.path.uab.edu/ (accessed on 20 September 2022). Asterisks designate statistically significant differences compared to normal samples; between groups (where accompanied by brackets) as analyzed by unpaired *t*-test with Welch’s correction or statistically significant linear trend between group means and left-to-right order (where accompanied by an arrow) as analyzed by one-way ANOVA test; *: *p* < 0.05, **: *p* < 0.01, ***: *p* < 0.001, ****: *p* < 0.0001. (**C**) Kaplan–Meier plots depicting the probability of overall survival in months in LUAD patients exhibiting high (red) or low (black) expression levels of *IL-37*, *SIGIRR*. Hazard ratio (HR), logrank *p*-values, number of patients with either high or low gene expression categorized also in those who survived for 50, 100, 150, 200, and 250 months are reported. Graphs were exported from www.kmplot.com [34].

### 3.4. Increased IL-37 Expression Is a Favorable Prognostic Factor for Overall Survival in LUAD Patients

We further searched for possible correlations between *IL-37* expression levels and survival rates in LUAD patients utilizing the Kaplan–Meier Plotter tool [34]. Analysis of RNA sequencing data from 504 individuals with LUAD tumors revealed significant differences in overall survival (OS) time between patients with tumors expressing high (n = 226) vs. low *IL-37* levels (n = 278) (*log* rank *p* = 0.021). (Figure 2C). Specifically, high *IL-37* expression increases the probability of survival at 250 months by 29% (hazard ratio (HR) = 0.71. 95% CI = 0.53–0.95) compared to the low *IL-37*-expressing group. Further, median survival time for the *IL-37* high expression cohort was 54.07 months, whereas for the low expression cohort, it was 39.9 months. In the case of *SIGIRR*, its high expression increases the probability of survival at 250 months by 35% compared to low expression of the gene (HR = 0.65 (0.49–0.88), *p* = 0.0041; n = 235 and n = 269 for patients with high and low expression, respectively). The median survival time for the *SIGIRR* high expression cohort was 55.1 months, whereas for the low expression cohort, it was 40.3 months. We also checked the prognostic potential of the mean expression of the two genes, which was found to be weaker than the levels of each individual gene (HR = 0.74 (0.55–1), *p* = 0.046; n = 167 and n = 337 for patients with high and low mean expression, respectively).

Lastly, we analyzed the prognostic potential of *IL-37* and *SIGIRR* in individual group biopsies of different grade (1–4), stage (1–4), low or high mutation burden, and neoantigen load. The results revealed that both genes exhibit differential patterns and the ability to predict LUAD OS in patients bearing biopsies of distinct histopathological characteristics. Statistics of the analysis can be found in Appendix A. A low sample size, though, did not allow us to perform a combinatorial analysis of the parameters. 

### 3.5. Correlation of IL-37 Expression with Cancer-Associated Genes in LUAD Tumors

The UALCAN portal [33] was used to explore possible linear association of *IL-37* mRNA levels with the expression of other genes in LUAD tumors. As shown in Figure 3A, there are 20 genes that are positively associated with *IL-37* expression, as revealed upon processing RNAseq data from TCGA biopsies and analysis with Pearson’s correlation test. In details, *IL-37* levels (log_2_TPM + 1) were positively associated with those of: *PRODH* (enzyme proline dehydrogenase), *HINF1A* (hepatocyte nuclear factor 1-alpha), *DPP4* (dipeptidyl-peptidase 4), *DGCR5* (DiGeorge syndrome critical region gene 5), *DUSP6* (dual-specificity phosphatase 6), *NMNAT2* (nicotinamide nucleotide adenylyltransferase 2), *HLF* (hepatic leukemia factor), *ADORA1* (adenosine A1 receptor), *SHF* (Src homology 2 domain containing F), *MFSD4* (major facilitator superfamily domain containing 4A), *CXCL14* (C-X-C motif chemokine ligand 14), *ITGA2* (integrin subunit alpha 2), *DGCR9* (DiGeorge syndrome critical region gene 9), *CEACAM2* (carcinoembryonic antigen-related cell adhesion molecule 2), *PLAT* (plasminogen activator, tissue type), *PPP1R1B* (protein phosphatase 1 regulatory inhibitor subunit 1B), *STK39* (serine/threonine kinase 39), *MUC1* (mucin 1, cell surface-associated), *DPY19L1* (dpy-19 like C-mannosyltransferase 1), and *CDC42EP1* (CDC42 effector protein 1) (*p* for all < 0.0001) (Figure 3B). No gene was found to be negatively associated with *IL-37* in LUAD tumors.

Special interest was set on the exploration of possible associations of IL-37 with certain immune-checkpoint (IC) molecules. Since the UALCAN portal did not provide data on correlations for the non-significant associations, we explored the correlation analysis tool in the TNM plotter [32]. Spearman’s *r* and *p*-values of linear associations with IL-37 were: *r* = 0.05 and *p* = 0.265 for PD-1 (or PDCD1; programmed cell death protein 1), *r* = 0.13, *p* = 0.0035 for PD-L1 (CD274), *r* = 0.12 and *p* = 0.0065 for CTLA-4 (cytotoxic T-lymphocyte-associated protein 4), *r* = 0.1 and *p* = 0.023 for VISTA (V-domain Ig suppressor of T cell activation or VSIR V-set immunoregulatory receptor), *r* = −0.08 and *p* = 0.0892 for LAG3 (lymphocyte activating 3), *r* = −0.19 and *p* = 0.00001 for TIM-3 (T cell membrane protein 3 or HAVCR2: hepatitis A virus cellular receptor 2).

STRING database [37] processing data from various depositories was utilized to pinpoint any potential cellular or molecular networks or biological processes that are regulated by the aforementioned set of positively correlated genes. “Regulation of leukocyte migration” and “regulation of response to external stimulus” were the two biological processes (Gene Ontology) enriched (false-discovery rate (FDR): 0.0032 for both; level of confidence for minimum required interaction score: 0.4). Since the activity of IL-37 lies mainly in the suppression of proinflammatory signaling factors such as mTOR, AKT, and PI3K, all involved in autophagy and key metabolic functions of immune and cancer cells [2], we further processed individual correlations with genes in these processes. In the aforementioned list, we pinpointed *HIF1A* as a key metabolic regulator, further implicated in key processes during cancer development and progression [38,39], and we also further explored for possible associations with OS of LUAD patients through the Kaplan–Meier Plotter tool [34]. No association was revealed when *HIF1A* was analyzed; however, the ratio of expression levels of *IL-37*-to-*HIF1A* was found to possess a prognostic potential in this cohort (HR = 0.67 (0.5–0.92), *p* = 0.012; low ratio samples n = 130, high expression samples n = 374). 

Linear correlations with SIGIRR’s expression levels were also explored. UALCAN portal [33] revealed 646 genes positively and 34 genes negatively associated with SIGIRR. The total list of the genes and the corresponding Pearson’s *r* values can be found in Appendix A. Gene-set enrichment analysis of SIGIRR-related genes via the STRING database [37] did not reveal any cancer/LUAD-related entrance in any of the STRING-connected repositories.

### 3.6. IL-37 Protein Expression Correlates with the Grade of LUAD Tumor

Following gene expression analysis, investigation of the protein expression pattern through the Pathology tool of the Human Protein Atlas website [31] initially revealed that both IL-37 and SIGIRR are expressed in lung tumor biopsies (Figure 4A). IL-37 protein levels were similar between 111 LUAD and paired non-LUAD specimens (median z-value (range): 0.11 (−1.02–1.16) vs. −0.043 (−1.02–1.16), *p* = 1.10 × 10^−1^) (Table 3, Figure 4B). However, protein levels were associated with tumor grade: the lowest levels were observed in grade 2 biopsies and the highest in grade 3, whereas intermediate levels in grade 1 biopsies led to non-cancerous tissues. Regarding SIGIRR protein expression, this was modestly upregulated in tumor tissues compared to normal ones (−0.25 (−1.14–0.38) vs. 0 (−1.91–2.26), *p =* 2.82 × 10^−2^) and was associated with tumor grade, since SIGIRR levels exhibited a linear decrease following the grade 1-to-3 order (*r*^2^ = 0.1634, *p* = 0.0001).

### 3.7. T-Lymphocytes and Macrophages of the Lung Express IL-37 and SIGIRR Genes

According to single-cell RNA sequencing (scRNAseq) data deposited and processed through the Single-Cell Atlas (of the Human Protein Atlas) [31], *IL-37* was found to be expressed by resident T lymphocytes and macrophages of the normal lung tissue (read counts normalized to transcripts per million protein coding genes (pTPM) = 2.1 and 1.2, respectively) (Appendix A). Moreover, *SIGIRR* was found to be expressed by immune cell populations, including T cells, granulocytes, and macrophages, but also by other resident cell types including alveolar cell types 1 and 2, fibroblasts, club ciliated cells, and endothelial cells. The highest *SIGIRR* expression was detected in T cells (pTPM = 115.2) and the lowest in macrophages (pTPM = 23.6). It is also noteworthy that, based on the the scRNAseq analysis, IL-37 protein expression levels in normal human lung cell subsets are relatively lower compared to those of SIGIRR. 

### 3.8. IL37 Gene Alterations Correlate with Differential Immune Cell Infiltration of the Lung Tumor

To investigate the effects of IL-37 gene mutations on immune cell infiltration in lung adenocarcinoma tumors, the “Mutation” module of the TIMER2.0 webserver was applied [36]. Our analysis indicates that tumors bearing non-synonymous, somatic mutations in the *IL-37* gene were characterized by significantly higher infiltration of CD4^+^ T lymphocytes (CYBERSORT project; log_2_fold-change = 2.105, Wilcoxon *p* = 0.004) and significantly lower infiltration of M2 macrophages (XCELL project; log_2_fold-change = −2.709, Wilcoxon *p* = 0.009) and neutrophils (MCPCOUNTER project; log_2_fold-change = −0.665, *p* = 0.031). The contribution of myeloid dendritic cells (mDCs) within tumor-association infiltration was of similar levels in patients with and without IL-37 non-synonymous mutations (log_2_fold-change = 1.171, *p* = 0.047) (Figure 5A).

### 3.9. IL37 Expression Levels Correlate with Infiltration Levels of Certain Immune Cell Subsets

Exploration through the “Gene” module of the TIMER2.0 webserver [36] revealed that *IL37* gene expression levels, as assessed in previous RNAseq experiments, were linearly correlated with certain immune cell populations infiltrating the LUAD tumor. More specifically, *IL37* expression levels (log_2_TPM) were positively associated with the infiltration rate of mDCs (XCELL project; Spearman’s rho = 0.42, *p* = 1.84 × 10^−22^) progenitors of granulocytes-monocytes (GMPs) (XCELL project; Spearman’s rho = 0.323, *p* = 2.07 × 10^−13^), activated mast cells (CIBERSORT-ABS project; Spearman’s rho = 0.303, *p* = 5.89 × 10^−12^), Tregs (QUANTISEC project; Spearman’s rho = 0.241, *p* = 5.70 × 10^−8^), and M2 macrophages (QUANTISEC project; Spearman’s rho = 0.279, *p* = 2.80 × 10^−10^) and negatively associated with the infiltration rate of MDSCs (TIDE project; Spearman’s rho = −0.292, *p* = 3.54 × 10^−11^) (Figure 5B); all correlations are summarized in Table 4.

### 3.10. IL-37 Signaling Shares Common Nodes with PD-1/PDL-1 and CTLA-4 Immune Checkpoint Pathways

Finally, analysis using the STRING database [37] processing data from the various depositories revealed that protein molecules involved in the IL-37 signaling pathway are also members of cancer-related pathways (Figure 6). Specifically, STAT3 is involved in “PD-L1 expression and PD-1 checkpoint pathway in cancer” (KEGG database) and “Cancer immunotherapy by PD-1 blockade” (WikiPathways), PTPN6 in “PD-L1 expression and PD-1 checkpoint pathway in cancer” (KEGG database)/“PD-1 signaling” (Reactome), and “Cancer immunotherapy by CTLA4 blockade” (WikiPathways), and PTPN11 is involved in all of the above. False discovery rates of the significance for each of the enhanced pathways are: 0.0015 for PD-L1 expression and PD-1 checkpoint pathway in cancer, 0.0029 for cancer immunotherapy by CTLA4 blockade, 0.0053 for cancer immunotherapy by PD-1 blockade, and 0.0081 for PD-1 signaling.

## 4. Discussion

Lung cancer is a leading cause of cancer death in both men and women, worldwide [40,41,42]; it is associated with almost three times the rate of deaths compared to prostate cancer in men and breast cancer in women. In 2022, 236.740 estimated new cases of lung cancer are expected to be diagnosed, and approximately 130.180 deaths are predicted to be recorded in the United States of America (USA) [43]. However, during the last 15 years, a steady decline in the incidence of new lung cancer diagnoses (2.8% in men and 1.4% in women) and deaths (50% and 67%, respectively) has been observed. This is probably attributed to recent advances in the management of patients, including the use of novel chemotherapeutic (cisplatin/pemetrexed or gemcitabine), molecular targeted (gefitinib), and/or immunotherapeutic (pembrolizumab, nivolumab, atezolizumab) agents, together with more personalized therapeutic approaches based on specific mutation patterns, such as those on *EGFR*, *ALK,* and *ROS-1* genes [44]. 

Lung adenocarcinoma (LUAD), which falls under the umbrella of non-small-cell lung cancer (NSCLC), represents about 40% of all lung cancer types [45], and even though there has been a significant decrease in its incidence and mortality rates, it remains the main cause of cancer death in the USA [45]. LUAD tumors that evolve primarily from mucosal glands are usually developed in the lung periphery but can also be found in scars or areas of chronic inflammation. The immune microenvironment exerts functions associated with the development and progression of the disease, as well as the response to therapy [40]. Alternatively activated MΦ (M2 type) and T cells, specifically resting memory CD4^+^, are the predominant populations surrounding LUAD tumors [46]. Further, high immune cell infiltration rates correlate with a better prognosis compared to lower ones [47,48] and are characterized by an increased incidence of naïve B cells, plasma cells, follicular helper T cells, and classical (M1 type) macrophages, as well as by a decreased prevalence of resting memory CD4^+^ T cells, monocytes, and resting dendritic cells (DCs) [48]. 

Recently, Zuo et al. developed the immune-cell characteristic score (ICCS) model, which is suggested for the facilitation of LUAD prognosis [49]. This model assesses the infiltration rate of six immune cell populations in LUAD biopsies (B cells, immature DCs, eosinophils, mast cells, granzyme K expressing CD8^+^ T cells, and Th2 cells) and independently predicts the overall survival (OS) of the patient. High infiltration of all the ICCS immune cell populations is associated with better prognosis, apart from that of Th2 cells, which indicate a poor outcome of the disease [49]. What is more, processing RNA-sequencing and clinical data from TCGA [50] and GEO databases [51] utilizing specific bioinformatics tools and algorithms has led to the construction of certain gene signatures with independent prognostic/predictive values for the survival of LUAD patients [47,48,52] or their response to immune-checkpoint inhibitors (ICIs) [53,54]. 

Focusing on the pivotal role of the tumor-immune microenvironment in LUAD development and progression, we aimed at the exploration of the expression patterns of *IL-37*, a novel cytokine with regulatory properties in LUAD tumors, using various bioinformatics tools and available databases. Based on our data, human LUAD tumors exhibit the highest gene-expression levels of the cytokine amongst various common human cancers, suggesting its possible pathophysiological involvement in this malignancy. Importantly, these are significantly increased in LUAD vs. non-LUAD tissues, suggesting a disease-specific involvement of IL-37 in the pathological lesion. Within LUAD, LBC-mucinous and mucinous-coloid histological subtypes are characterized by the highest *IL-37* levels; however, the small number of specimens assessed for each subtype does not allow for the extraction of safe conclusions regarding the histological-specific distribution of the cytokine.

It is of note that IL-37 expression exhibits a lung cancer stage-specific pattern: there is a trend of steady decline in its levels from earlier to later stages. This observation provides initial evidence for a possible prognostic value for *IL-37* expression patterns, but it also implies a potential pathogenetic role of this regulatory cytokine in the LUAD inflammatory lesion. However, there are clearly several unresolved questions regarding the molecular mechanisms underlying this differential expression profile, such as whether aberrant *IL-37* regulation is a cause or consequence of lung cancer development and progression. Therefore, it is of outmost importance to pinpoint the exact cellular source(s) of IL-37 that are responsible for the variability in its distribution (LUAD cells and/or immune cell subsets) before making any assumptions about the link between molecular/cellular pathogenetic mechanisms and the mRNA signature of *IL-37* in LUAD malignancies. Even though most of the current knowledge advocates towards a possible protective role of this cytokine in human malignancies, the enhanced expression in early-stage lung tumors, independently from the exact cellular source(s), may indicate a cellular response to promote anti-tumor immunity. Consequently, the observed gradual decrease in IL-37 expression during tumor progression and advanced lung cancer stages suggests that this anti-tumor effect could probably be attenuated. In contrast to the differential expression pattern of *IL-37* levels between LUAD and non-LUAD biopsies, this does not apply for its receptor *SIGIRR*. The receptor’s gene expression levels between the groups are similar; nevertheless, within LUAD, the mixed, LBC-mucinous, and mucinous-coloid subtypes exhibit a tendency to express higher *SIGIRR* mRNA levels, possibly associated with the aforementioned enhanced *IL-37* levels in the same subtypes. What is of interest, though, is the sharp trend of decrease observed following the stage 1-to-stage 4 order, also matching with the corresponding *IL-37* signature. 

Our results also reveal the prognostic potential of tumor-expressed *IL-37* and *SIGIRR* patterns. Patients whose biopsies exhibit high *IL-37* or *SIGIRR* mRNA levels have a better prognosis for OS compared to those with low levels. This, together with the stage-specific expression patterns described above, suggests that lower *IL-37* and *SIGIRR* gene expression could be associated with more advanced LUAD cases, of high grade/stage and/or metastatic status linked to poorer prognosis. Yet, *SIGIRR* seems to be an even stronger OS predictor, which is in accordance with the more significant, compared to *IL-37*, correlations with stage and metastasis status described herein. 

*IL-37* expression levels were found to be positively correlated with a 20-gene signature in LUAD biopsies. Most of these genes have an already established association with lung cancer and specifically with LUAD. Increased levels of *CXCL14* mRNA have been detected in LUAD biopsies with a micropapillary pattern [55], and smoking-induced *CXCL14* expression in the human airway epithelium has been implicated in chronic obstructive pulmonary disease (COPD)-mediated lung cancer development [56]. Similarly, *NMNAT2* expression was found to be increased in LUAD specimens and correlated negatively with OS of patients, whereas the DGUOK-NMNAT2-NAD+ axis was suggested as a potential therapeutic target for the disease [57]. Increased expression levels of *HLF,* shown to promote cell-cycle progression in various cancers [58], *ITGA2*, *MUC1*, and *DPY19L1* have also been proposed to confer prognostic value for the survival of patients suffering from LUAD malignancies [59,60,61,62,63,64]. Additionally, *DUSP6*/*MKP3* has been designated as a tumor suppressor phosphatase implicated in LUAD and cancer types, whereas several studies have revealed the clinical relevance of its expression patterns in lung cancer. In addition, various *DUSP6*/*MKP3*-associated SNPs have been linked with the response to chemoradiation therapy [65,66,67,68,69]. Similarly, *PPP1R1B* has been shown to interfere with the response to molecular targeted therapy in *EGFR*-mutated LUAD [70], and *SHP2* has been associated with *MET*-mutated NSCLC [71].

Moreover, additional *IL-37*-correlated genes have been described to be implicated in NSCLC pathogenesis and progression. *PLAT* has been reported to inhibit apoptosis in NSCLC cells, and its knockdown augments the therapeutic efficacy of gefitinib [72]. *PRODH* has been involved in NSCLC metastasis as shown, both in vitro and in vivo [73,74], and *ADORA1*, which is highly expressed in *EGFR*-mutant NSCLC biopsies [75], has been associated involved in tumor-immune evasion in NSCLC xenograft models [76]. *STK39* and the lncRNA *DGCR5* have been proposed as critical molecules for the regulation of the growth, migration, and invasion of NSCLC tumors [74,77,78,79]. *DGCR5* has been specifically implicated in the tumor progression of LUAD through the inhibition of hsa-mir-22-3p [80]. Regarding the remaining genes that exhibit an *IL-37*-correlated expression pattern, there is evidence for their association with the development of human cancers and for their potential to serve as disease biomarkers. This is the case with *HIF1A*, the most pivotal gene regulating metabolic pathways related to hypoxia, which is further implicated in proliferation, energy metabolism, invasion, and metastasis in a series of human cancers, and has been viewed as a highly promising therapeutic target [38,39]. Indeed, there is evidence that expression levels of *HIF1A* by tumor cells have a diagnostic and prognostic significance among different histological types of lung cancer [81,82]. Interestingly, in our study, the ratio of *IL-37*-to-*HIF1A* expression levels was found to have a favorable prognostic potential in LUAD patients. Further, *MFSD4* is considered to be a tumor-suppressor gene and a biomarker for hepatic metastasis in gastric cancer patients [83], as well as a diagnostic marker of esophageal carcinoma [83]. In addition, the lncRNA *DGCR9* has been reported as a potential tumor neoantigen [84], with a possible pathogenetic role in gastric cancer [85]. As for *CDC42EP1*, it was very recently described that certain gene mutations drive the development of parathyroid and oral tongue squamous cell carcinomas [86,87]. Lastly, *DPP-4* has been reported to possess a deleterious role and potential to be used as a biomarker in respiratory diseases, such as lung cancer, asthma, and chronic obstructive pulmonary disease (COPD) [88,89,90]. 

Regarding IL-37 protein expression, this was found to be similar for both proteins between LUAD and non-LUAD specimens. As with mRNA expression, its levels were associated with the tumor grade. It is essential to comment that the webportal used provided the opportunity to analyze the gene expression levels in regard to the stage of the tumor, as well as the protein levels with regard to its grade. Given the differences among these terms and scales, one can speculate that the significant trend of decrease in *SIGIRR* gene expression levels is pathophysiologically connected to the trend in protein levels, both associated with more severe disease. In the case of *IL-37*, things seem to be more complicated: biopsies of intermediate severity (Grade 2) express the highest gene expression but the lowest protein levels, respectively, compared to the rest of the subgroups in each classification. Further exploration for these seemingly opposite regressions needs to be performed. 

By taking advantage of the TIMER2.0 webserver, we had the opportunity to search for possible correlations between *IL-37* gene alterations and infiltration rates of various immune cell types in LUAD. Interestingly, it was revealed that tumors with non-synonymous, somatic mutations of *IL-37* are characterized by a higher infiltration of CD4^+^ T-cells and a lower infiltration of M2 ΜΦ and neutrophils. Although the differences are statistically significant, due to the low number of samples in the mutated arm (n = 5), the conclusions cannot be confidently evaluated. What is more, even though these mutations are expected to alter protein function, assumptions about the pathophysiological links of the above relationships, if not falsely positive, could be made upon determining their cell-specific distribution, as well as the activation status of the corresponding immunocytes. It is essential to confirm whether *IL-37* aberrations indeed affect the infiltrating incidence and/or function, especially of M2 ΜΦ and neutrophils, which dominate the myeloid-landscape of LUAD tumors and are of vital importance for their growth and metastatic potential [91]. 

Supplementing these observations, the *IL-37* gene expression pattern was found to be significantly associated with the infiltration rate of certain immune cell types. Myeloid dendritic cells (mDCs), which showed the most significant correlation, are known to support protective anti-tumor immunity in lung cancer, while being subjected to suppression mediated by cancer cells via different mechanisms [92,93]. In agreement, there was also a positive association between the percentage (%) of GMPs, which produce DCs and macrophages, and they have also been shown to be involved in LUAD immune cell infiltration [94]. Moreover, *IL-37* mRNA levels positively correlated with the percentage (%) of activated mast cells, which have been assigned as predictors of improved OS and PFS in NSCLC [49,95]. In contrast, *IL-37* expression was negatively correlated with the percentage (%) of infiltrating MDSCs, which are pivotal immunosuppressive partners and key targets for immunotherapy in lung cancer [96]. The above observations support the notion that IL-37 may exert tumor-protective immune functions. However, it should also be noted that our analyses revealed some, at first sight, conflicting evidence: *IL-37* expression levels correlated positively with the rate of infiltration by regulatory T cells, as well as M2 MΦ, that have been reported to exert pro-tumoral and anti-tumor immunity actions in LUAD [97,98]. The molecular and cellular networks responsible for these phenotypes need further investigation. 

Finally, pathway analysis revealed that IL-37-signaling mediators, such as STAT3, are crucial partners of PD-1, PD-L1, and CTLA-4 pathways, providing hints for an interfering role of this cytokine in the immune-checkpoint blockade of anti-tumor immune responses. 

Taken together, our data highlight the prognostic and diagnostic potential of *IL-37* mRNA levels in LUAD and provide evidence for its involvement in molecular networks and cellular distributions reported to play pivotal roles in LUAD tumorigenesis and progression. What is more, the interplay with the SIGIRR receptor, as well as its possible disturbance and/or independence, as possibly reflected by their differential expression pattern, points toward a crucial role in the differential disease phenotype, which could be further exploited as a potential immune-checkpoint therapeutic target. Our results are in agreement with previous studies supporting the implication of this cytokine and its receptor in the anti-tumor cytotoxic [24,99,100] and anti-angiogenic responses [101], as well as anti-invasion/metastatic processes in NSCLC [102,103]. Specifically, for LUAD, a study on patients’ samples showed that the loss of or reduced IL-37 expression in the tumor correlates with metastasis development [23]. The protective effect of IL-37 has also been shown in other lung diseases such as idiopathic pulmonary fibrosis (IPF) [104]. 

However, the current study is limited in certain parameters, such as the fact that it is based on clinical samples analyzed through omics approaches. Therefore, it is essential that the resulted data are further validated in certain large patient cohorts of interest using specifically designed, targeted assays (including RT-qPCR). A detailed exploration of possible associations with clinical, histopathological, laboratory, and therapeutic parameters needs to be performed to empower the capacity of the *IL-37* to be conceivably used as a LUAD biomarker. For a better understanding of the pathogenetic implications of the cytokine, serial biopsies of LUAD specimens could be used to monitor its differential expression profile throughout the course of the disease and/or in association with certain treatment strategies. Further, peripheral blood vs. tumor samples should be comparably processed to explore the inflamed-tissue or peripheral LUAD specific distribution of *IL-37* as a demand of local or systemic immunoregulation, as previously described in other immune-mediated inflammatory disorders [105,106]. Moreover, it is important to further deepen our study and investigate the likely differential distribution of each of the five IL-37 isoforms [2], both in mRNA and protein levels. The analysis of the total mRNA and protein isoforms in -omics assays could hide specific patterns of certain variants, which also need to be further checked as to whether they exert similar or different functions in the LUAD microenvironment. The different isoforms bear different exons that have been implicated in extracellular and intranuclear activities of the cytokine, thus mediating different signaling regulations [2]. Additionally, and in combination with the above, it is of outmost importance to pinpoint the exact cellular source(s) of IL-37 and its receptor, to fully understand the effect of their aberrations in the intercellular responses within the LUAD microenvironment. Single-cell analyses supported their expression by lung tissue-resident T cells and ΜΦ, but relevant study should also be applied in LUAD tumor biopsies to explore the possible inducible expression of IL-37 isoforms and SIGIRR by malignant epithelial and/or tumor-infiltrating immune cell subsets. Towards that direction, our preliminary data support the fact that both IL-37 mRNA and protein are expressed by A549 human lung adenocarcinoma cells, as attested by a specifically developed RT-qPCR assay and flow cytometry (Appendix A). SIGIRR expression was not detected in those cells by the aforementioned approaches. 

Despite its limitations, the current study clearly supports the crucial involvement of IL-37 in LUAD pathogenesis and monitoring. Moreover, it highlights the plausible necessity for further investigation through mechanistic studies at the molecular and cellular level and validation in experimental models as well as in well-defined patient cohorts, in order to fully elucidate the exact role of this cytokine and to further exploit its potential for the improvement of LUAD patients’ personalized management. 

## Figures and Tables

**Figure 1 biomedicines-10-03037-f001:**
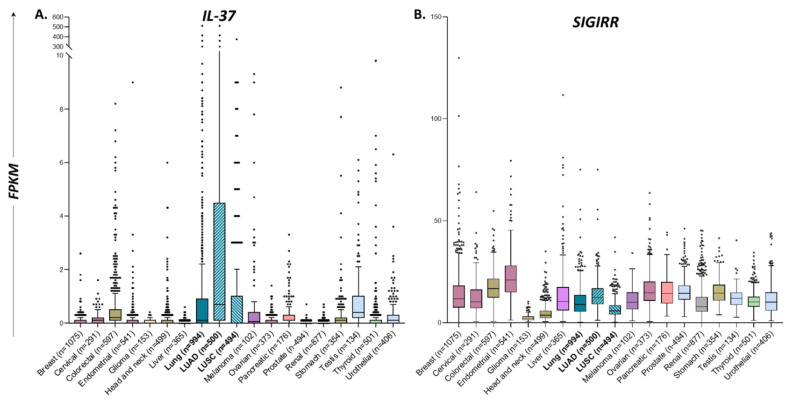
Box and Tukey whiskers diagrams showing the differential distribution of *IL-37* (**A**) and *SIGIRR* (**B**) expression levels (FPKMs), as analyzed by RNA sequencing in human tumor samples of various cancer types. Especially for lung cancer, samples were separately analyzed as lung adenocarcinoma (LUAD) or squamous carcinoma (LUSC). Data were obtained from www.proteinatlas.org [31] (accessed on 24 August 2022) and further processed for statistical analysis. Left and right sides of the boxes correspond to the lower and upper quartiles; the box covers the interquartile interval including 50% of the data. The vertical line in the box represents the median. Whiskers outside the box expand from the minimum to the lower quartile and from the upper quartile to the maximum of data range. Each dot represents the value of a single sample.

**Figure 3 biomedicines-10-03037-f003:**
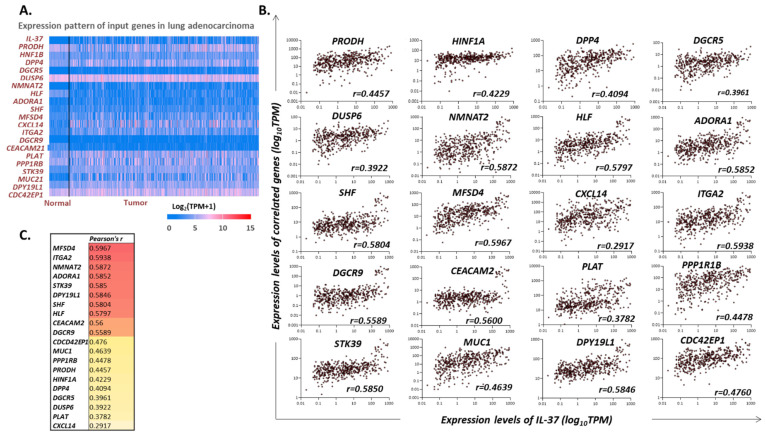
(**A**) Heatmap depicting the relative expression levels (log_2_TPM + 1) of *IL-37* and twenty IL-37-correlated genes in normal and LUAD biopsies as analyzed by RNAseq. (**B**) Dot plot diagrams showing the correlation between *IL-37* and each of the twenty significantly positively correlated genes. No negative associations were detected. Pearson’s *r* values are reported in each case. All *p*-values were <0.0001. Figures were exported from http://ualcan.path.uab.edu/ [33] (accessed on 10 September 2022). (**C**) Heatmap depicting the grade of association between the expression levels of each of the twenty *IL-37*-positively correlated genes and the levels of *IL-37*. Pearson’s *r* values are reported.

**Figure 4 biomedicines-10-03037-f004:**
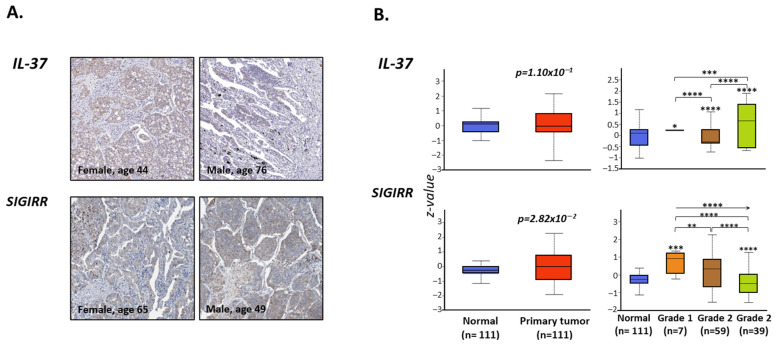
(**A**) Immunohistochemical staining of IL-37 and SIGIRR proteins in lung biopsies of LUAD patients. Pictures were obtained from www.proteinatlas.org [35] (accessed on 20 September 2022). (**B**) Box and Tukey whiskers diagrams depicting the differential protein expression levels of IL-37 and SIGIRR (z-values) in LUAD primary tumors (n = 111) vs. normal tissues (n = 111) and among LUAD tumors of different grade (1 to 3). Figures and graphs were exported from http://ualcan.path.uab.edu/ [33] (accessed on 20 September 2022). *: *p* < 0.05, **: *p* < 0.01, ***: *p* < 0.001, ****: *p* < 0.0001.

**Figure 5 biomedicines-10-03037-f005:**
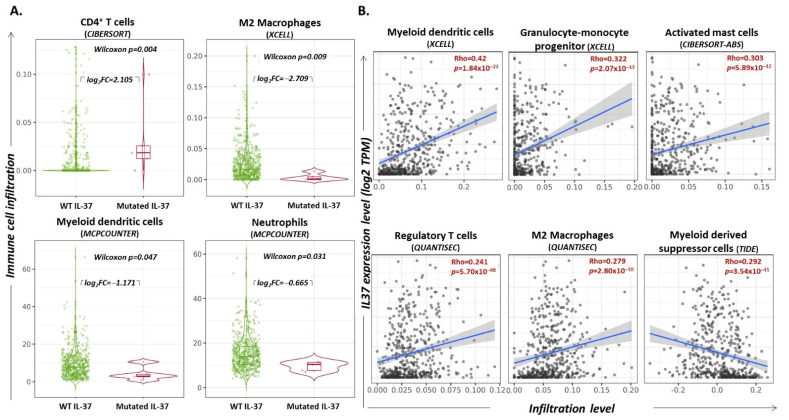
(**A**) Violin plots depicting the distribution of infiltrated T cells, M2 macrophages, myeloid dendritic cells, and neutrophils in LUAD tumors without versus with mutation on *IL-37*. Wilcoxon *p*-values and log_2_ (fold-changes, FC) are reported. (**Β**) Scatter plot diagrams depicting the linear association between levels of *IL-37* gene expression (log_2_TPM; y-axis) and infiltration of the tumor by certain cell subsets (x-axis). Spearman’s Rho and *p*-values are reported. Data were filtered for tumor purity. Graphs were exported from http://timer.cistrome.org/ [36] (accessed on 1 October 2022).

**Figure 6 biomedicines-10-03037-f006:**
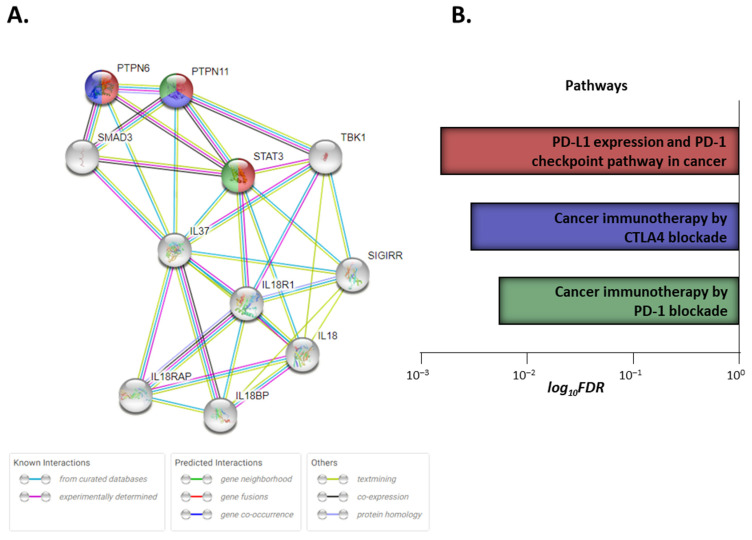
Network of interactions (exported by STRING portal [37]) (**A**) among proteins involved in IL-37 signaling and other proteins implicated in PD-1/PD-L1 and CTLA-4 immune-checkpoint pathways (**B**), as attested using the STRING (Search Tool for the Retrieval of Interacting Genes/Proteins). The above pathways share common nodes (each color corresponds to different pathway; nodes marked with more than one color belong to equal number of pathways). Edges represent protein–protein associations: known interactions, predicted interactions, or other associations. Level of confidence for minimum required interaction score was 0.4. Data were obtained from https://string-db.org [37] (accessed on 3 October 2022).

**Table 1 biomedicines-10-03037-t001:** *IL-37* and *SIGIRR* mRNA expression levels (FPMKs) in various human cancer types. Data analyzed were obtained from www.proteinatlas.org [31] (accessed on 24 August 2022). *p*-Values compared to total lung or LUAD specimens using non-parametric Mann-Whitney U test are reported.

Human Cancer Type	No of Patients	Gene Expression Levels			
		*IL-37*		*SIGIRR*	
		Mean FPKM ± SD	*p*-Value Compared to Lung/LUAD	Mean FPKM ± SD	*p*-Value Compared to Lung/LUAD
Breast	1075	0.07 ± 0.58	<0.0001/<0.0001	14.04 ± 10.16	<0.0001/NS
Cervical	291	0.17 ± 0.19	<0.0001/<0.0001	12.77 ± 8.29	<0.0001/0.0006
Colorectal	597	0.68 ± 2.38	0.09/<0.0001	17.64 ± 7.15	<0.0001/<0.0001
Endometrial	541	0.19 ± 1.45	<0.0001/<0.0001	22.52 ± 10.64	<0.0001/<0.0001
Glioma	153	0.05 ± 0.07	<0.0001/<0.0001	2.56 ± 1.68	<0.0001/<0.0001
Head-and-neck	499	0.18 ± 0.49	<0.0001/<0.0001	5.04 ± 4.21	<0.0001/<0.0001
Liver	365	0.03 ± 0.07	<0.0001/<0.0001	14.02 ± 12.76	<0.0001/0.0008
Lung	994	6.59 ± 31.88	NA/NA	10.30 ± 6.78	NA/NA
LUAD	500	12.85 ± 44.05	NA/NA	13.64 ± 7.03	NA/NA
LUSC	494	2.56 ± 17.89	NA/0.0001	6.92 ± 4.46	NA/<0.0001
Melanoma	102	0.83 ± 2.11	0.0002/<0.0001	10.96 ± 6.44	NS/<0.0001
Ovarian	373	0.10 ± 0.15	<0.0001/<0.0001	16.70 ± 8.41	<0.0001/<0.0001
Pancreatic	176	0.34 ± 0.54	NS/<0.0001	15.54 ± 7.74	<0.0001/0.005
Prostate	494	0.01 ± 0.05	<0.0001/<0.0001	15.57 ± 6.26	<0.0001/<0.0001
Renal	877	0.02 ± 0.06	<0.0001/<0.0001	9.94 ± 6.93	0.05/<0.0001
Stomach	354	0.35 ± 1.21	0.0001/<0.0001	15.17 ± 5.94	<0.0001/<0.0001
Testis	134	1.08 ± 2.27	NS/NS	12.25 ± 5.31	<0.0001/0.08
Thyroid	501	0.25 ± 1.24	<0.0001/<0.0001	11.02 ± 4.66	<0.0001/<0.0001
Urothelial	406	0.67 ± 8.58	<0.0001/<0.0001	11.40 ± 7.3	0.006/<0.0001

NA: not applicable, NS: non-significant.

**Table 2 biomedicines-10-03037-t002:** Differential distribution of *IL-37* and *SIGIRR* mRNA levels in LUAD vs. non-LUAD tissues or among LUAD tumors of different histological type or stage. Number of patients in each group, the median levels of expression and their range (where available), fold-change over non-LUAD/normal samples, *p*-values, and the webtool used for the analysis (TNMplot [32], UACLAN [33]) are reported.

	*IL-37*				*SIGIRR*				
Non-LUAD vs. LUAD									
	n	Levels (Median; Range)	Fold-Change	*p*-Value	n	Levels (Median; Range)	Fold-Change	*p*-Value	Platform
Non-LUAD	486	1			486	959.5			TNMplot
LUAD	524	11	12	3.83 × 10^−59^	524	811	0.85	5.63 × 10^−5^	TNMplot
Normal	59	0.08 (0–0.38)			59	35.06 (17.01–55.55)			UACLAN
Primary tumor	515	0.43 (0–19.6)	5.38	5.97 × 10^−11^	515	38.71 (3.84–84.81)	1.11	9.20 × 10^−7^	UACLAN
Tumors vs. adjacent Normal									
Adjacent normal	57	1			57	713			TNMplot
Tumor	57	11	11	3.81 × 10^−8^	57	749	1.05	6.28 × 10^−1^	TNMplot
LUAD histological subtype									
Normal	59	0.08 (0–0.38)			59	35.06 (17.01–55.55)			UACLAN
NOS	320	0.42 (0–16.99)	5.25	NS	320	37.08 (3.84–84.81)	1.06	NS	UACLAN
Mixed	107	0.7 (0–42.55)	5.6	NS	107	42.19 (9.77–71.23)	1.20	0.0003	UACLAN
Clear cell	2	22.40 (0–44.79)	280	NS	2	30.54 (29.11–31.98)	0.87	0.0022	UACLAN
LBC-Nonmucinous	19	1.08 (0–14.61)	13.5	NS	19	31.81 (24.68–59.76)	0.90	NS	UACLAN
Solid pattern predominant	5	0.10 (0–3.13)	1.25	NS	5	35.81 (27.55–62.29)	1.02	NS	UACLAN
Acinar	18	0.34 (0–9.66)	4.25	NS	18	48.39 (20.35–122.59)	1.38	NS	UACLAN
LBC-Mucinous	5	1.07 (0.20–1.96)	13.38	0.0072	5	52.32 (37.87–60.69)	1.49	0.0008	UACLAN
Mucinous colloid	10	0.26 (0–0.76)	3.25	0.0153	10	47.72 (35.68–70.80)	1.36	0.0011	UACLAN
Papillary	23	0.30 (0–19.55)	3.75	NS	23	43.01 (23.36–70.09)	1.23	NS	UACLAN
Mucinous	2	0.09 (0–0.17)	1.13	NS	2	56.86 (52.91–60.81)	1.62	NS	UACLAN
Micropapillary	3	11.24 (3.76–31.37)	140.05	NS	3	23.67 (21.42–73.27)	0.67	NS	UACLAN
Signet ring	1	0.82 (0.82–0.82)	10.25	NA	1	45.86 (45.86–45.86)	1.30	NS	UACLAN
Cancer stage									
Normal	59	0.08 (0–0.38)			59	35.06 (17.01–55.55)			UACLAN
Stage 1	277	28 (0–23.62)	7.63	NS	277	42.53 (3.97–89.73)	1.21	<0.0001	UACLAN
Stage 2	125	0.37 (0–13.55)	4.63	NS	125	36.16 (12.59–75.62)	1.03	NS	UACLAN
Stage 3	85	0.17 (0–19.55)	2.13	NS	85	33.03 (3.84–74.01)	0.94	NS	UACLAN
Stage 4	28	0.35 (0–6.17)	4.38	NS	28	25.65 (7.76–69.02)	0.73	NS	UACLAN
*Linear trend Stage 1* *→* *4*				NS				<0.0001	

NOS: not otherwise specified, LBC: lung bronchioloalveolar carcinoma, NS: non-significant, NA: not applicable. Other between-two-groups comparisons; for *IL-37*; NOS vs. mixed: NS, NOS vs. clear cell: NA, NOS vs. LBC-nonmucinous: NS, NOS vs. solid pattern predominant: NS, NOS vs. acinar: *p* = 0.0192, NOS vs. mucinous: *p* = 0.000003, NOS vs. mucinous colloid: NA, NOS vs. papillary: NS, NOS vs mucinous: *p* = 0.000003, NOS vs. micropapillary: NS, NOS vs. signet ring: NA, mixed vs. clear cell: NA, mixed vs. LBC-nonmucinous: NS, mixed vs. solid pattern predominant: NS, mixed vs. acinar: *p* = 0.011, mixed vs. LBC-mucinous: *p* = 0.0006, mixed vs. mucinous colloid: NA, mixed vs. papillary: NS, mixed vs. mucinous: NA, mixed vs. micropapillary: NS, mixed vs. signet ring: NA, clear cell vs. LBC-nonmucinous: NA, clear cell vs. solid pattern predominant: NA, clear cell vs. acinar: NA, clear cell vs. LBC-mucinous: NA, clear cell vs. mucinous colloid: NA, clear cell vs. papillary: NA, clear cell vs. mucinous: NA, clear cell vs. micropapillary: NA, clear cell vs. signet ring: NA, LBC-nonmucinous vs. solid pattern predominant: NS, LBC-nonmucinous vs. acinar: NS, LBC-nonmucinous vs. LBC-mucinous: NS, LBC-nonmucinous vs. mucinous colloid: NA, LBC-nonmucinous vs. papillary: NS, LBC-nonmucinous vs. papillary: NS, LBC-nonmucinous vs. nucinous:, LBC-nonmucinous vs. micropapillary: NS, LBC-nonmucinous vs. signet ring: NA, solid pattern predominant vs. acinar: NS, solid pattern predominant vs. LBC-mucinous: NS, solid pattern predominant vs. mucinous colloid: NS, solid pattern predominant vs. papillary: NS, solid pattern predominant vs. mucinous: NS, solid pattern predominant vs. micropapillary: NS, solid pattern predominant vs. signet ring: NA, acinar vs. LBC-mucinous: NS, acinar vs. mucinous colloid: NA, acinar vs. papillary: NS, acinar vs. mucinous: NA, acinar vs. micropapillary: NS, acinar vs. signet ring: NA, LBC-mucinous vs. mucinous colloid: NA, LBC-mucinous vs. papillary: NS, LBC-mucinous vs. mucinous: NS, LBC-mucinous vs. micropapillary: NS, LBC-mucinous vs. signet ring: NA, for *SIGIRR*; NOS vs. mixed: NS, NOS vs. clear cell: NS, NOS vs. LBC-nonmucinous: NS, NOS vs. solid pattern predominant: NS, NOS vs. acinar: *p* = 0.0066, NOS vs. mucinous: NS, NOS vs. mucinous colloid: NS, NOS vs. papillary: NS, NOS vs mucinous: NS, NOS vs. micropapillary: NS, NOS vs. signet ring: NA, mixed vs. clear cell: NS, mixed vs. LBC-nonmucinous: *p* = 0.0422, mixed vs. solid pattern predominant: NS, mixed vs. acinar: *p* = 0.0307, mixed vs. LBC-mucinous: NS, mixed vs. mucinous colloid: NS, mixed vs. papillary: NS, mixed vs. mucinous: NS, mixed vs. micropapillary: NS, mixed vs. signet ring: NA, clear cell vs. LBC-nonmucinous: NS, clear cell vs. solid pattern predominant: NS, clear cell vs. acinar: NS, clear cell vs. LBC-mucinous: NS, clear cell vs. mucinous colloid: NS, clear cell vs. papillary: NS, clear cell vs. mucinous: *p* = 0.0245, clear cell vs. micropapillary: NS, clear cell vs. signet ring: NA, LBC-nonmucinous vs. solid pattern predominant: NS, LBC-nonmucinous vs. acinar: *p* = 0.0140, LBC-nonmucinous vs. LBC-mucinous: NS, LBC-nonmucinous vs. mucinous colloid: *p* = 0.0414, LBC-nonmucinous vs. papillary: NS, LBC-nonmucinous vs. papillary: NS, LBC-nonmucinous vs. mucinous, LBC-nonmucinous vs. micropapillary: NS, LBC-nonmucinous vs. signet ring: NA, solid pattern predominant vs. acinar: NS, solid pattern predominant vs. LBC-mucinous: NS, solid pattern predominant vs. mucinous colloid: NS, solid pattern predominant vs. papillary: NS, solid pattern predominant vs. mucinous: NS, solid pattern predominant vs. micropapillary: NS, solid pattern predominant vs. signet ring: NA, acinar vs. LBC-mucinous: NS, acinar vs. mucinous colloid: NA, acinar vs. papillary: NS, acinar vs. mucinous: NA, acinar vs. micropapillary: NS, acinar vs. signet ring: NA, LBC-mucinous vs. mucinous colloid: NS, LBC-mucinous vs. papillary: NS, LBC-mucinous vs. mucinous: NS, LBC-mucinous vs. micropapillary: NS, LBC-mucinous vs. signet ring: NA.

**Table 3 biomedicines-10-03037-t003:** Differential distribution of IL-37 and SIGIRR protein levels in LUAD vs. normal tissues or among LUAD tumors of different grades. Number of patients in each group, the median levels of expression, and their range as obtained from UALCAN portal [33], as well as *p*-values of statistical differences are reported.

	*IL-37*			*SIGIRR*			
Non-LUAD vs. LUAD							
	n	Levels (Median; Range)	*p*-Value	n	Levels (Median; Range)	*p*-Value	Platform
Normal	111	0.11 (−1.02–1.16)		111	−0.25 (−1.14–0.38)		UACLAN
Primary tumor	111	−0.043 (−1.02–1.16)	NS	111	0 (−1.91–2.26)	2.82 × 10^−2^	UACLAN
Grade							
Normal	111	0.11 (−1.02–1.16)		111	−0.25 (−1.14–0.38)		UACLAN
Grade 1	7	0.22 (0.21–0.24)	0.0313	7	0.93 (−0.23–1.35)	0.001	UACLAN
Grade 2	59	−0.28 (−0.75–1.07)	0.0351	59	0.35 (−1.55–2.26)	NS	UACLAN
Grade 3	39	0.66 (−0.68–1.90)	<0.0001	39	−0.48 (−1.57–1.26)	<0.0001	UACLAN
Linear trend Grade 1 → 3			NS			0.0001	

**Table 4 biomedicines-10-03037-t004:** Associations between the levels of expression of *IL-37* and those of infiltration by certain immune cell types in LUAD tumors, as attested in 515 specimens. Spearman’s *r* values are reported. For significant associations, *p*-values are also mentioned. Tumor purity filter was applied. Data were obtained from www.timer.cistrome.org, (accessed on 24 August 2022) [36].

	EPIC	TIMER	MCP-COUNTER	QUANTISEQ	CIBERSORT	CIBERSORT-ABS	XCELL	TIDE
CD8^+^ T cell	−0.007; NS	0.032; NS	−0.029; NS	−0.002; NS	−0.187; 3 × 10^−5^	−0.051; NS	0.026; NS	NA
Naïve	NA	NA	NA	NA	NA	NA	−0.03	NA
Memory	NA	NA	NA	NA	NA	NA	0.069	NA
Central memory	NA	NA	NA	NA	NA	NA	−0.034; NS	NA
CD4^+^ T cell	0.006; NS	0.06; NS		−0.168 *; 2 × 10^−4^	NA	NA	0.088 *; NS	NA
Naïve	NA	NA	NA	NA	0.019; NS	0.019; NS	0.126; 5 × 10^−3^	NA
Memory	NA	NA	NA	NA	NA	NA	−0.019; NS	NA
Central memory	NA	NA	NA	NA	NA	NA	0.04; NS	NA
Effector memory	NA	NA	NA	NA	NA	NA	0.384; 8 × 10^−20^	NA
Memory activated	NA	NA	NA	NA	−0.163; 3 × 10^−4^	−0.158; 4 × 10^−4^	NA	NA
Memory resting	NA	NA	NA	NA	0.208; 3 × 10^−6^	0.259; 5 × 10^−9^	NA	NA
Th1	NA	NA	NA	NA	NA	NA	−0.103; 3 × 10^−2^	NA
Th2	NA	NA	NA	NA	NA	NA	−0.323; 2 × 10^−13^	NA
Tregs	NA	NA	NA	0.241; 6 × 10^−8^	0.119; 6 × 10^−3^	0.167; 2 × 10^−4^	NA	NA
B cell	−0.017; NS	0.126; 5 × 10^−3^	0.052; NS	−0.026; NS	NA	NA	0.066; NS	NA
Naïve	NA	NA	NA	NA	−0.103; 2 × 10^−2^	−0.061; NS	0.047; NS	NA
Memory	0.26; NS	NA	NA	NA	0.202; 6 × 10^−6^	0.204; 5 × 10^−6^	−0.031; NS	NA
Neutrophil	NA	0.033; NS	0.145; 1 × 10^−3^	0.311; 2 × 10^−12^	−0.032; NS	−0.011; NS	−0.146; 1 × 10^−3^	NA
Monocyte	NA	NA	−0.056; NS	−0.029; NS	0.207; 3 × 10^−6^	0.226; 4 × 10^−7^	0.26; 4 × 10^−9^	NA
Macrophage	0.137; 2 × 10^−3^	−0.01; NS	−0.056; NS	NA	−0.147; 1 × 10^−3^	−0.113; 1 × 10^−2^	0.12; 8 × 10^−3^	NA
M1	NA	NA	NA	0.388; 4 × 10^−19^	−0.109; 2 × 10^−2^	−0.018; NS	−0.014; NS	NA
M2	NA	NA	NA	0.279; 3 × 10^−10^	0.21; 3 × 10^−6^	0.251; 2 × 10^−8^	0.166; 2 × 10^−4^	0.047; NS
Dendritic cell								
Myeloid	NA	NA	0.406; 5 × 10^−21^	−0.141; NS	NA	NA	0.42; 2 × 10^−22^	0.307; 4 × 10^−12^
Myeloid activated	NA	NA	NA	NA	0.101; 2 × 10^−2^	0.138; 2 × 10^−3^	0.316; 7 × 10^−13^	NA
Myeloid resting	NA	NA	NA	NA	NA	NA	NA	NA
Plasmacytoid	NA	NA	NA	NA	NA	NA	−0.124; 6 × 10^−3^	NA
Natural killer	−0.086; NS	NA	−0.239; 8 × 10^−8^	0.16; 7 × 10^−4^	NA	NA	−0.018; NS	NA
Activated	NA	NA	NA	NA	−0.188; 3 × 10^−5^	−0.064; NS	NA	NA
Resting	NA	NA	NA	NA	−0.078; NS	−0.077; NS	NA	NA
Mast cell	NA	NA	NA	NA	NA	NA	0.201; 7 × 10^−6^	NA
Activated	NA	NA	NA	NA	0.272; 9 × 10^−10^	0.303; 6 × 10^−12^	NA	NA
Resting	NA	NA	NA	NA	−0.159; 4 × 10^−4^	−0.138; 2 × 10^−3^	NA	NA
Cancer-associated fibroblast	0.065; NS	NA	0.045; NS	NA	NA	NA	0.211; 2 × 10^−6^	0.063; NS
Common lymphoid progenitor	NA	NA	NA	NA	NA	NA	−0.14; 2 × 10^−3^	NA
Common myeloid progenitor	NA	NA	NA	NA	NA	NA	0.092; NS	NA
Endothelial cell	−0.007	NA	0.109; 2 × 10^−3^	NA	NA	NA	−0.062; NS	NA
Eosinophil	NA	NA	NA	NA	0.067	NA	0.067; NS	0.005; NS
Granulocyte-lymphocyte progenitor	NA	NA	NA	NA	NA	NA	0.323; 2 × 10^−13^	NA
Hematopoieitic stem cell	NA	NA	NA	NA	NA	NA	0.159; 4 × 10^−4^	NA
follicular helper T cell	NA	NA	NA	NA	−0.187; NS	NA	NA	NA
γδ T cell	NA	NA	NA	NA	−0.021; NS	−0.02; NS	0.025; NS	NA
NK Τ cell	NA	NA	NA	NA	NA	NA	0.43; NS	NA
MDSCs	NA	NA	NA	NA	NA	NA	NA	−0.292; 4 × 10^−11^

* non-regulatory, NA: not available, NS: non-significant, Tregs: regulatory T cells, MDSCs: myeloid-derived suppressor cells.

## Data Availability

Not applicable.

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
