# Peer review of "Aberrant Expression and Prognostic Potential of IL-37 in Human Lung Adenocarcinoma"

_biomedicines, 2022, doi:10.3390/biomedicines10123037_

Round 1

Reviewer 1 Report

Christodoulou et al. investigate the prognostic and/or diagnostic potential of interleukin-37 (IL-37), a rather recently identified cytokine with dual immunosuppressive and/or immune-stimulating role, and its receptor SIGIRR in cancer. The findings of the study support that IL-37 may has a utility as a prognostic marker of lung adenocarcinoma (LUAD) and its staging, as well as overall survival of patients.

This is a well-designed and presented study. Its major disadvantage, as also the authors state, is that is based exclusively in the analysis of publicly available databases using a series of bioinformatics tools, without further experimental verification.

The presentation of the data and the writing of the manuscript are exceptional. The following minor issues should be clarified:

1  1. As also authors analyze in the introduction IL-37 has a dual role (suppression and/or stimulation) depending on the microenvironment and tissue of action. Thus, the statement that “this cytokine possesses a pivotal role in preventing excessive inflammation in various human disorders including autoimmune diseases, inflammatory systemic conditions, infections and cancer” in the first paragraph, although true, is somehow confusing, considering that mostly inflammatory properties of IL-37 in cancer are described in the fourth paragraph.

2       2. The findings of the study suggest a prognostic, but not a diagnostic, role of IL-37 and its receptor in LUAD, since their expression levels were similar to those of normal samples. So claims of its potential diagnostic role should be omitted from discussion section.

3       3. The values presented in Figure 1 and Table 1 are not similar. In this context, SIGIRR levels in Figure 1B are elevated mainly in endometrium and to a lesser extend in breast, colorectal, ovarian, pancreatic, cervical etc. cancers, whereas in Table 1 higher values are presented in breast and liver. The authors should examine the reasons of this discrepancy. Furthermore, the plots in Figure 1 should be described in more detail (what squares, bars, dots represent).  

Author Response

Comments and Suggestions for Authors

Christodoulou et al. investigate the prognostic and/or diagnostic potential of interleukin-37 (IL-37), a rather recently identified cytokine with dual immunosuppressive and/or immune-stimulating role, and its receptor SIGIRR in cancer. The findings of the study support that IL-37 may has a utility as a prognostic marker of lung adenocarcinoma (LUAD) and its staging, as well as overall survival of patients.

This is a well-designed and presented study. Its major disadvantage, as also the authors state, is that is based exclusively in the analysis of publicly available databases using a series of bioinformatics tools, without further experimental verification.

The presentation of the data and the writing of the manuscript are exceptional. The following minor issues should be clarified:

Point 1: As also authors analyze in the introduction IL-37 has a dual role (suppression and/or stimulation) depending on the microenvironment and tissue of action. Thus, the statement that “this cytokine possesses a pivotal role in preventing excessive inflammation in various human disorders including autoimmune diseases, inflammatory systemic conditions, infections and cancer” in the first paragraph, although true, is somehow confusing, considering that mostly inflammatory properties of IL-37 in cancer are described in the fourth paragraph.

Response 1: We thank the reviewer for pointing out this inconsistency. We agree that this may cause confusion to the readers, so we amended the sentence as follows: “Thus, this cytokine possesses a pivotal role in inflammation related to the pathophysiology of various human disorders including autoimmune diseases, inflammatory systemic conditions, infections and cancer [1].” [line 1 of the revised manuscript]

Point 2: The findings of the study suggest a prognostic, but not a diagnostic, role of IL-37 and its receptor in LUAD, since their expression levels were similar to those of normal samples. So claims of its potential diagnostic role should be omitted from discussion section.

Response 2: We thank the reviewer for meticulously checking our manuscript. We agree that neither IL-37 nor SIGIRR proteins have a diagnostic potential in LUAD, since they are of similar levels in LUAD vs non-LUAD biopsies. However, regarding mRNA levels, as described in Fig 2A and Table 2, IL-37 is expressed in significantly higher levels:

  1. in LUAD primary tumors vs normal lung biopsies (data exported from TNMplotter: 12-fold increase, p=3.83x10-59, data exported from UALCAN: 5.38-fold increase, p=5.97x10-11) and
  2. in LUAD primary tumors vs paired adjacent normal tissue (data exported from TNMplotter: 11-fold increase, p=3.81x10-8)

Indeed, there is no difference between LUAD and non-LUAD tissues regarding SIGIRR mRNA levels (even though there are statistically significant p values, fold-changes do not designate a biologically meaningful alteration in this parameter).

Following reviewer’s comment, and to be accurate, we amended the lines 638-639 of the revised version as: “the prognostic and diagnostic potential of IL-37 mRNA levels”, adding the words mRNA levels to state the gene and not protein expression levels. As double-checked there is no other comment on the diagnostic potential of the two molecules in the Discussion section.

Point 3: The values presented in Figure 1 and Table 1 are not similar. In this context, SIGIRR levels in Figure 1B are elevated mainly in endometrium and to a lesser extend in breast, colorectal, ovarian, pancreatic, cervical etc. cancers, whereas in Table 1 higher values are presented in breast and liver. The authors should examine the reasons of this discrepancy. Furthermore, the plots in Figure 1 should be described in more detail (what squares, bars, dots represent). 

Response 3: Once more, we thank the reviewer for the meticulous review of our work and we apologize for any confusion. Indeed, we inadvertently added wrong numbers in the “SIGIRR- Mean FPKM±SD” column of the table. We have now corrected them and we also made some corrections to typos (adding two instead of one decimal places) in corresponding IL-37 values.

We also included additional information about the representation of the data in the Figure 1, in the corresponding legend, as suggested by the reviewer: “…and further processed for statistical analysis. Left and right sides of the boxes correspond to the lower and upper quartiles, the box covers the interquartile interval including 50% of the data. The vertical line in the box represents the median. Whiskers outside the box expand from the minimum to the lower quartile and from the upper quartile to the maximum of data range. Each dot represents the value of a single sample.” [lines 182-186 of the revised manuscript].

Reviewer 2 Report

In this manuscript Christodoulou et. al. characterize the potential of IL-37 and SIGIRR as prognostic biomarkers for tumour development. Although the concept of IL-37 to act as potential biomarker in cancer is not new, since several studies have suggested its potential for prognosis in HCC, CRC, AML and other types of malignancies, the authors demonstrate in this manuscript its ability to serve as biomarker for human lung adenocarcinoma as well, that to my knowledge has not been reported before. Thus, I believe that the manuscript is of potential interest to the reads of the journal. 

My specific comments to the authors are:

1) The authors should cite and comment on the following manuscript doi: 10.1002/cam4.1663 by Bing Zhu et al, that identifies the prognostic significance of nomograms integrating IL-37 expression, neutrophil level, and MMR status in patients with colorectal cancer.

2) The authors in their introduction state that the studies on the role of IL-37 in cancer are limited. A search in the recent literature I feel that proves that this statement is not definite, since there are extensive papers characterizing the role of IL-37 in diverse types of tumors. The authors should reshape that comment.

3) The research team in Figure 1 and 2 utilized the Tissue Atlas tool of the Human Protein Atlas database to investigate the expression levels of IL-37 and its receptor SIGIRR in tumor biopsies of 17 different cancer types. Their analysis revealed that IL-37 expression significantly differs in various types of human cancer, with lung cancers and especially LUAD exhibiting the highest levels. Based on these results the authors subsequently focus on LUAD and explore its correlation with disease progression and OS. The authors need to better justify and substantiate their choice since low expression of IL-37 can also correlate with low overall survival as in the case of other reports in the literature for CRC.

4) The results of the manuscript and the conclusions drawn could benefit from analyzing peripheral blood levels of IL-37 and correlating to disease progression and OS in the cancer types investigated by the team. 

5) In Figure 2C the authors have separately analyzed the expression of IL-37 and SIGIRR and concluded that patients with high IL-37 expression or high SIGIRR expression have a greater survival probability. What about patients that exhibit high SIGIRR along with concomitant high IL-37 expression? What is their correlation to OS and compared to other expression pattern combinations?

6) Since the activity of IL-37 lies mainly on the suppression of proinflammatory signaling factors such as mTOR, AKT and PI3K all involved in autophagy and key metabolic functions of immune and cancer cells, it would be interesting to investigate IL-37 expression and correlate it to key autophagic or metabolic proteins and OS.  

Author Response

Comments and Suggestions for Authors

In this manuscript Christodoulou et. al. characterize the potential of IL-37 and SIGIRR as prognostic biomarkers for tumour development. Although the concept of IL-37 to act as potential biomarker in cancer is not new, since several studies have suggested its potential for prognosis in HCC, CRC, AML and other types of malignancies, the authors demonstrate in this manuscript its ability to serve as biomarker for human lung adenocarcinoma as well, that to my knowledge has not been reported before. Thus, I believe that the manuscript is of potential interest to the reads of the journal.

My specific comments to the authors are:

Point 1: The authors should cite and comment on the following manuscript doi: 10.1002/cam4.1663 by Bing Zhu et al, that identifies the prognostic significance of nomograms integrating IL-37 expression, neutrophil level, and MMR status in patients with colorectal cancer.

Response 1: This is indeed a very interesting article supporting the independent prognostic potential of IL-37 in CRC, and we thank the reviewer for suggesting it. We added this citation and related information in lines 121-128 of the revised manuscript.

Point 2: The authors in their introduction state that the studies on the role of IL-37 in cancer are limited. A search in the recent literature I feel that proves that this statement is not definite, since there are extensive papers characterizing the role of IL-37 in diverse types of tumors. The authors should reshape that comment.

Response 2: We thank the reviewer for spotting this. Indeed, there is a significant amount of ever-increasing evidence on the role of IL-37 in cancer. We amended the corresponding sentence in the introduction as: “Evidence on the role of IL-37 in cancer, emerging during the last years, support…” [line 73 of the revised manuscript]

Point 3: The research team in Figure 1 and 2 utilized the Tissue Atlas tool of the Human Protein Atlas database to investigate the expression levels of IL-37 and its receptor SIGIRR in tumor biopsies of 17 different cancer types. Their analysis revealed that IL-37 expression significantly differs in various types of human cancer, with lung cancers and especially LUAD exhibiting the highest levels. Based on these results the authors subsequently focus on LUAD and explore its correlation with disease progression and OS. The authors need to better justify and substantiate their choice since low expression of IL-37 can also correlate with low overall survival as in the case of other reports in the literature for CRC. 

Response 3: We thank the reviewer for giving us the opportunity to justify more our selection on LUAD. As mentioned also in the manuscript, the first criterion for picking up the most “promising” type of cancer to study was the disease-specific differential expression profile of IL-37, and indeed this was most prominent in the case of LUAD. However, we also performed a preliminary search in the rest of the bioinformatics tools and databases (TNMplot, KMplotter, UALCAN), to identify cancer types with IL-37 differential expression between normal and tumor samples and/or paired tumor and adjacent normal samples, correlation between IL-37 expression levels with survival rates etc. The first most promising cancer type was LUAD followed by pancreatic and bladder cancers. Even though we did consider previous references in the bibliography, our selection was based primarily on the evidence obtained from large cohorts such as those processed by the aforementioned portals. This was both because we are interested in analyzing as larger number of data as possible for a higher soundness of the results, and also because we aimed at contributing to the specific Special Issue of the Journal regarding omics approaches for the identification of novel biomarkers and disease targets. Apart from LUAD, we have also developed similar studies for pancreatic and bladder cancers that are now running.

Regarding CRC, the approach described above did not reveal any dramatic changes in IL-37 expression. In details, analysis through the TNM plotter revealed no significant change between colon adenocarcinoma tumors and paired adjacent normal samples (n=41; p=0.07, fold change=0.67), nor between colon adenocarcinoma (n=469) and non-cancerous colon biopsies (n=315) (p=0.01 but marginal fold-change=1.50). Similarly, UALCAN portal revealed a 0.92-fold change between primary colon adenocarcinoma biopsies (n=286) and normal biopsies (n=41) (p=3x10-5). Regarding, correlation with survival rates, UALCAN portal data support no correlation between IL-37 expression levels and OS (p=0.64; high expression n=71, low expression n=208), while no data were available to be processed in KMplotter).

Point 4: The results of the manuscript and the conclusions drawn could benefit from analyzing peripheral blood levels of IL-37 and correlating to disease progression and OS in the cancer types investigated by the team. 

Response 4: We thank the reviewer for mentioning this. Unfortunately, no data on peripheral blood samples from LUAD patients and data on disease parameters and survival rates in the same patients are available in public databases (including GEO2R), that could be analyzed and included in this study. However, indeed, this is the next step we are now moving forward in our ongoing study: comparative analysis of paired samples (LUAD tumor biopsies and peripheral blood samples) and exploration of possible association with histopathological, clinical, laboratory characteristics, progression of the disease and survival rates. We are also interested in analysing the exact peripheral blood and TME cell source(s) responsible for the differential pattern. This is a project following our first observational data reported in this manuscript, that to our opinion could stand as an independent project and hopefully form a future publication. We are happy that the reviewer suggests the necessity also for this study.

Point 5: In Figure 2C the authors have separately analyzed the expression of IL-37 and SIGIRR and concluded that patients with high IL-37 expression or high SIGIRR expression have a greater survival probability. What about patients that exhibit high SIGIRR along with concomitant high IL-37 expression? What is their correlation to OS and compared to other expression pattern combinations?

Response 5: This is a very interesting comment. Towards this approach, we utilized the option of KMplotter tool to analyze the mean expression of two genes (in our case IL-37 and SIGIRR). However, the significance of each individual gene (for IL-37: HR=0.71, p=0.021, for SIGIRR: HR=0.65, p=0.0041) was higher than their mean expression (for the ratio: HR=0.74, p=0.046). We added these findings in the related paragraph in the Results section [lines 291-294 of the revised manuscript]: “We also checked the prognostic potential of the mean expression of the two genes, which was found to be weaker than the levels of each individual gene [HR=0.74 (0.55-1), p=0.046; n=167 and n=337 for patients with high and low mean expression, respectively]”.

Point 6: Since the activity of IL-37 lies mainly on the suppression of proinflammatory signaling factors such as mTOR, AKT and PI3K all involved in autophagy and key metabolic functions of immune and cancer cells, it would be interesting to investigate IL-37 expression and correlate it to key autophagic or metabolic proteins and OS. 

Response 6: We thank the reviewer for this great suggestion. To find possible correlations of IL-37 expression with genes/proteins implicated in autophagy and metabolic functions of cells in the tumor microenvironment of LUAD patients, we searched in the list of associated genes revealed by processing RNA sequencing data through the UALCAN portal (Figure 3). We detected positive association with HIF1A gene (Pearson’s r=0.4229), which as discussed also in the Discussion section: “…the most pivotal gene regulating metabolic pathways related to hypoxia, which is further implicated in proliferation, energy metabolism, invasion, and metastasis in a se-ries of human cancers, and has been viewed as a highly promising therapeutic target [36, 37]. Indeed, there are evidence that expression levels of HIF1A by tumor cells, have a diagnostic and prognostic significance among different histological types of lung cancer [81, 82]” [lines 577-581 of the revised manuscript]. Upon, reviewer’s comment we searched for the possible prognostic potential of that gene in LUAD through the KMplotter tool. Nevertheless, this revealed no potential for LUAD OS prognosis by this gene [HR=1.23 (0.92-1.65), p=0.16; low expression samples n=255, high expression samples n=249]. The same applied when the mean expression of IL-37 and HIF1A was assessed: [HR=1.25 (0.92-1.7), p=0.15; low expression samples n=196, high expression samples n=308]. However, a significance was revealed when the ratio of the two genes was accessed: [HR=0.67 (0.5-0.92), p=0.012; low ratio samples n=130, high expression samples n=374]. We added this information to the corresponding paragraph of the results section, lines 344-354 of the revised manuscript: “Since the activity of IL-37 lies mainly on the suppression of proinflammatory signaling factors such as mTOR, AKT and PI3K all involved in autophagy and key metabolic functions of immune and cancer cells [2], we further processed individual correlations with genes in these processes. In the aforementioned list, we pinpointed HIF1A a key metabolic regulator, further implicated in key processes during cancer development and progression [36, 37], and we further explored for possible associations also with OS of LUAD patients, through the Kaplan-Meier Plotter tool [34. No association was revealed when HIF1A was analyzed; however, the ratio of expression levels of IL-37-to-HIF1A was found to possess a prognostic potential in this cohort [HR=0.67 (0.5-0.92), p=0.012; low ratio samples n=130, high expression samples n=374]”. We also commended on this result in the discussion section lines 582-583 of the revised manuscript: “Interestingly, in our study the ratio of IL-37-to-HIF1A expression levels was found to have a favourable prognostic potential in LUAD patients.”

Unfortunately, the levels of no autophagy-related genes were detected to be associated with those of IL-37 in LUAD tumors through these analyses.

Reviewer 3 Report

In legend to Figure 3, the relative expression of IL-37 seems to be calculated as log10TPM+1, while in the Figure 3A the heat-map units are log2TPM+1. Please, clarify it.  

In line 765 "availa ble" should be written as "available"

Author Response

Comments and Suggestions for Authors

Point 1: In legend to Figure 3, the relative expression of IL-37 seems to be calculated as log10TPM+1, while in the Figure 3A the heat-map units are log2TPM+1. Please, clarify it.

Response 1: We thank the reviewer for this comment and we apologize for any confusion. Corrected. The values are depicted in log2TPM+1, so we corrected the legend of the figure [line 332 of the revised manuscript]

Point 2: In line 765 "availa ble" should be written as "available"

Response 2: We thank the reviewer for the comment as well. Corrected. [line 812 of the revised manuscript]

Reviewer 4 Report

This study aimed to investigate the prognostic and diagnostic value of IL-37 in LUAD through multiple online databases. Although this is an interesting study, it could not be accepted for publication in this journal. There are my resasons:

1. All results are obtained from online databases (TCGA data), and there is no external or in-house cohort validation. How about the acutal mRNA and protein expression level of IL-37 and its receptors on LUAD cell lines and lung tissues? There are only correlation analysis, thus the reliability is poor based on these findings.

2. Now that the authors aimed to explore its prognostic value. Although survival curves showed that there was different prognosis between high- and log-IL37 groups, the results were obtained from log-rank test. How about its prognostic value when incorpated with oher clinical variables in the multivariable analysis?

3. The diagnostic value of single gene is wo weak. How about the diagnotis ability of IL-37? How about its AUC value in ROC curve?

4. Different online tools used different data format of IL-37, such as FPKM and  TPM. It should be unified.

5. Most importantly, there is no any cell lines or human sample experimental validations.

Author Response

Comments and Suggestions for Authors

This study aimed to investigate the prognostic and diagnostic value of IL-37 in LUAD through multiple online databases. Although this is an interesting study, it could not be accepted for publication in this journal. There are my reasons:

Point 1: All results are obtained from online databases (TCGA data), and there is no external or in-house cohort validation. How about the acutal mRNA and protein expression level of IL-37 and its receptors on LUAD cell lines and lung tissues? There are only correlation analysis, thus the reliability is poor based on these findings.

Response 1: We thank the reviewer for mentioning this. Indeed, as mentioned also in the Discussion section, it is essential that the data of this study are further validated in certain, large patient cohorts of interest, using specifically designed, targeted assays (including RT-qPCR). This should be accompanied by detailed exploration of possible associations with clinical, histopathological, laboratory and therapeutic parameters to empower the capacity of the IL-37 to be conceivably used as a LUAD biomarker. In the Discussion section, other extensions of the study are also analysed. Actually, these are parts of our ongoing study using lung cancerous and non-cancerous biopsies, peripheral blood samples and plasma specimens, heading towards validating and further investigating the LUAD-specific expression pattern of IL-37.

However, in this manuscript which was intended to be submitted for consideration for possible publication in this specific Special Issue of Biomedicines that welcomes submissions on omics approaches towards the identification of biomarkers and therapeutic targets, we included a series of results from a detailed approach through various omics tools to collect sound data that support the potential of this cytokine as a biomarker for LUAD. We feel that the number of samples analysed and the use of multiple tools ensure the validity of the data, serving the scope of the special issue.

(See also Response 5) However, upon reviewer’s comment, we added preliminary data on the expression of this cytokine in A549 human adenocarcinoma cells as Supplementary Figure 2 [lines 676-680 and 816-819 of the revised manuscript]. 

Point 2: Now that the authors aimed to explore its prognostic value. Although survival curves showed that there was different prognosis between high- and log-IL37 groups, the results were obtained from log-rank test. How about its prognostic value when incorpated with oher clinical variables in the multivariable analysis?

Response 2: This is a very good point. In fact, multivariate analysis is one of the endpoints of our ongoing study on real LUAD biopsies. Unfortunately, neither KMplotter nor UALCAN tools, give us the option of multivariate analysis, so we could not perform it. Taking advantage from reviewer’s comment and the options provided by the KMplotter tool however, we analyzed the prognostic potential of IL-37 and SIGIRR expression levels in individual groups biopsies of different grade (1-4), stage (1-4), low or high mutation burden and neoantigen load. Corresponding data have been added in new Supplementary Table 1 [line 299]. Simulating multivariate analysis, we tried to do combinatorial processing of the data related to IL-37 or SIGIRR prognostic value, but the low sample size did not allow to do so. We added relevant information in lines 295-300 of the revised manuscript: “Lastly, we analyzed the prognostic potential of IL-37 and SIGIRR in individual groups biopsies of different grade (1-4), stage (1-4), low or high mutation burden and neoantigen load. Results revealed that both genes exhibit differential patterns and ability to predict LUAD OS in patients bearing biopsies of distinct histopathological characteristics. Statistics of the analysis can be found in Supplementary Table 2. Low sample size though, did not allow us to perform combinatorial analysis of the parameters”.

Point 3: The diagnostic value of single gene is wo weak. How about the diagnotis ability of IL-37? How about its AUC value in ROC curve?

Response 3: We really apologize but we are not sure we understood this comment by the reviewer. We haven’t used/presented any ROC curve, so an AUC could apply. We have present survival data using Kaplan-Meier plots. We kindly ask the reviewer to clarify their comment, and we will be happy to address it.. 

Point 4: Different online tools used different data format of IL-37, such as FPKM and TPM. It should be unified.

Response 4: Unfortunately, samples’ data processed by the online tools are already analyzed and expressed as relative quantification values of genes; the users (we, in this case) do not have access to raw data (eg bam/sam, fastq files) to redo the analysis, in order to have unified values. However, TPM and FPKM are similar; they only differ in the order of operations through the pipeline until the last step of assessment of relative quantification value. Data within the same experiment need to be expressed in exactly the same module, but comparative data among experiments can safely be expressed by similar quantification measurements, such as TPM, FPKM, RPKM. We hope we addressed reviewer’s concern.

Point 5: Most importantly, there is no any cell lines or human sample experimental validations.

Response 5: We thank the reviewer for this comment. Even though the aim of the study was the inclusion of results revealed upon detailed analysis of data deposited in a series of bioinformatics tools, to support the potential of IL-37 as a biomarker in LUAD and fit with the scope of this Special Issue, upon reviewer’s comment we added preliminary data on the expression of this cytokine in A549 human adenocarcinoma cells as Supplementary Figure 2 [lines 676-680 and 816-819 of the revised manuscript]. The expression of IL-37 and SIGIRR as well as their functions in human lung adenocarcinoma cell lines and biopsies, are in the scope of our ongoing study.

Round 2

Reviewer 4 Report

Now, this manuscript could be accepted for publication.